# Learning Decentralized Partially Observable Mean Field Control for Artificial Collective Behavior

**Kai Cui, Sascha Hauck, Christian Fabian, Heinz Koeppl**
Dept. of Electrical Engineering and Information Technology, Technische Universität Darmstadt
`{kai.cui, heinz.koeppl}@tu-darmstadt.de`

## Abstract

Recent reinforcement learning (RL) methods have achieved success in various domains. However, multi-agent RL (MARL) remains a challenge in terms of decentralization, partial observability and scalability to many agents. Meanwhile, collective behavior requires resolution of the aforementioned challenges, and remains of importance to many state-of-the-art applications such as active matter physics, self-organizing systems, opinion dynamics, and biological or robotic swarms. Here, MARL via mean field control (MFC) offers a potential solution to scalability, but fails to consider decentralized and partially observable systems. In this paper, we enable decentralized behavior of agents under partial information by proposing novel models for decentralized partially observable MFC (Dec-POMFC), a broad class of problems with permutation-invariant agents allowing for reduction to tractable single-agent Markov decision processes (MDP) with single-agent RL solution. We provide rigorous theoretical results, including a dynamic programming principle, together with optimality guarantees for Dec-POMFC solutions applied to finite swarms of interest. Algorithmically, we propose Dec-POMFC-based policy gradient methods for MARL via centralized training and decentralized execution, together with policy gradient approximation guarantees. In addition, we improve upon state-of-the-art histogram-based MFC by kernel methods, which is of separate interest also for fully observable MFC. We evaluate numerically on representative collective behavior tasks such as adapted Kuramoto and Vicsek swarming models, being on par with state-of-the-art MARL. Overall, our framework takes a step towards RL-based engineering of artificial collective behavior via MFC.

## 1 Introduction

Reinforcement learning (RL) and multi-agent RL (MARL) has found success in varied domains with few agents, including e.g. robotics (Polydoros & Nalpantidis, 2017), language models (Ouyang et al., 2022) or transportation (Haydari & Yılmaz, 2020). However, tractability issues remain for systems with many agents, especially under partial observability (Zhang et al., 2021b). Here, specialized approaches give tractable solutions, e.g. via factorizations (Qu et al., 2020; Zhang et al., 2021a). We propose a general, tractable approach for a broad range of decentralized, partially observable systems.

**Collective behavior & partial observability.** Of practical interest is the design of simple local interaction rules to fulfill global, cooperative objectives by emergence of global behavior (Vicsek & Zafeiris, 2012). For example, intelligent self-organizing robotic swarms provide many applications such as farming, and general design frameworks remains elusive (Hrabia et al., 2018; Schranz et al., 2021). Other domains include group decision-making and opinion dynamics (Zha et al., 2020), biomolecular self-assembly (Yin et al., 2008), and active matter (Cichos et al., 2020; Kruk et al., 2020), e.g. nano-particles (Nasiri & Liebchen, 2022) or microswimmers (Narinder et al., 2018). Overall, there is a need for scalable MARL under decentralization and partial information.

**Scalable and partially observable MARL.** Despite its many applications, decentralized cooperative control remains a difficult problem even in MARL (Zhang et al., 2021b), especially if coupled

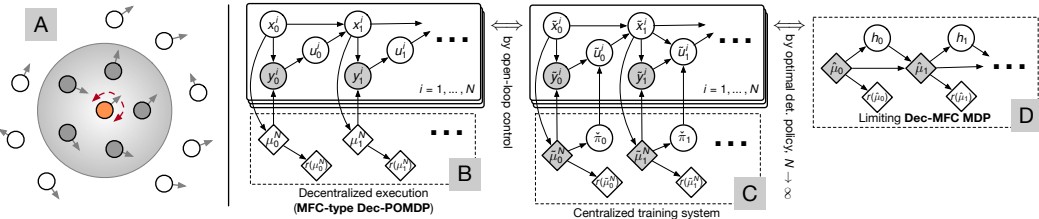

Figure 1: A: Partially-observable Vicsek problem: agents must align headings (arrows), but observe only partial information (e.g. heading distribution in grey circle for orange agent). B: The decentralized model as a graphical model (grey: observed variables). C: In centralized training, we also observe the mean field, guiding the learning of upper-level actions $\breve{\pi}$. D: The solved limiting MDP.

with the simultaneous requirement of scalability. Recent scalable MARL methods include graphical decompositions (Qu et al., 2020; Zhang et al., 2021a) amongst others (Zhang et al., 2021b). However, most remain limited to full observability (Zhang et al., 2021a). One line of algorithms applies pairwise mean field (MF) approximations over neighbors (Yang et al., 2018), which has yielded decentralized, partially observable extensions (Subramanian et al., 2021; 2022). Relatedly, MARL based on mean field games (MFG, non-cooperative) and mean field control (MFC, cooperative) focus on a broad class of systems with many exchangeable agents. While the theory for MFG is developed (Huang et al., 2006; Şen & Caines, 2019; Saldi et al., 2019), to the best of our knowledge, neither MFC-based MARL algorithms nor discrete-time MFC have been proposed under *partial information and decentralization*, except in special linear-quadratic cases (Tottori & Kobayashi, 2022; Wang et al., 2021). Further, MFGs have been useful for analyzing emergence of collective behavior (Perrin et al., 2021; Carmona et al., 2022), but less for "engineering" collective behavior to achieve global objectives as in MFC, which is our focus. This is in contrast to *rational, selfish* agents, as a decomposition of global objectives into per-agent rewards is non-trivial (Waelchli et al., 2023; Kwon et al., 2023). Beyond scalability to many agents, general MFC for MARL is also not yet scalable to *high-dimensional* state-actions due to discretization of the simplex (Carmona et al., 2019b; Gu et al., 2021), except in linear-quadratic models (Fu et al., 2019; Carmona et al., 2019a). Instead, we consider general discrete-time MFC and scale to higher dimensions via kernels. We note that our model has a similar flavor to TD-POMDPs (Witwicki & Durfee, 2010), as the MF also abstracts influence from all other agents. However, TD-POMDP addresses different types of problems, as it considers local per-agent states, while the MF is both globally shared and influenced by all agents.

**Our contribution.** A *tractable* framework for *cooperative* control, that can handle *decentralized, partially observable* systems, is missing. By the preceding motivation, we propose such a framework as illustrated in Figure 1. Our contributions may be summarized as (i) proposing the first discrete-time MFC model with decentralized and partially observing agents; (ii) providing accompanying approximation theorems, reformulations to a tractable single-agent Markov decision process (MDP), and novel optimality results over equi-Lipschitz policies; (iii) establishing a MARL algorithm with policy gradient guarantees; and (iv) presenting kernel-based MFC parametrizations of separate interest for general, higher-dimensional MFC. The algorithm is verified on classical collective swarming behavior models, and compared against standard MARL. Overall, our framework steps toward tractable RL-based engineering of artificial collective behavior for large-scale multi-agent systems.

## 2 Decentralized Partially Observable MFC

In this section, we introduce the motivating finite MFC-type decentralized partially observable control problem, as a special case of cooperative, general decentralized partially observable Markov decision processes (Dec-POMDPs (Bernstein et al., 2002; Oliehoek & Amato, 2016)). We then proceed to simplify in three steps of (i) taking the infinite-agent limit, (ii) relaxing partial observability during training, and (iii) correlating agent actions during training, in order to arrive at a tractable MDP with optimality guarantees, see also Figures 1 and 2. Proofs are found in Appendices D–S.

In a nutshell, Dec-POMDPs are hard, and hence we *reformulate* into the Dec-POMFC, for which we develop a new theory for optimality of Dec-POMFC solutions in the finite Dec-POMDP. The

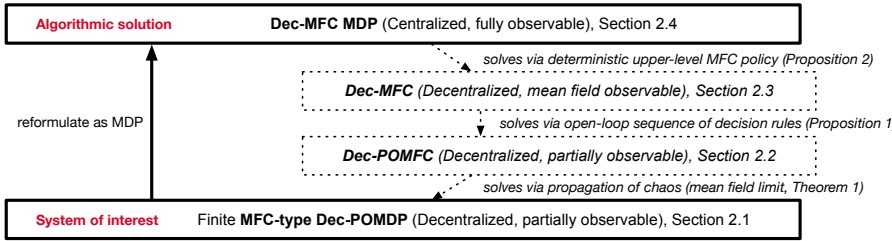

Figure 2: Three steps of approximation (mean field limit, open-loop control, and MDP reformulation) allow us to reformulate the broad class of MFC-type Dec-POMDP to a tractable Dec-MFC MDP.

solution of Dec-POMFC itself also remains hard, because its MDP is not just continuous, but *infinite-dimensional* for continuous state-actions. The MDP is later addressed in Section 3 by (i) kernel parametrizations and (ii) approximate policy gradients on the finite Dec-POMDP (Theorem 3).

## 2.1 MFC-TYPE COOPERATIVE MULTI-AGENT CONTROL

To begin, we define the finite Dec-POMDP of interest, which is assumed to be MFC-type. In other words, (i) agents are **permutation invariant**, i.e. only the overall distribution of agent states matters, and (ii) agents observe only part of the system. We assume agents $i \in [N] := \{1, \ldots, N\}$ endowed with random states $x_t^i$, observations $y_t^i$ and actions $u_t^i$ at times $t \in \mathcal{T} := \mathbb{N}$ from compact metric state, observation and action spaces $\mathcal{X}, \mathcal{Y}, \mathcal{U}$ (finite or continuous). Agents depend on other agents only via the empirical **mean field** $\mu_t^N := \frac{1}{N} \sum_{i \in [N]} \delta_{x_t^i}$. Policies are memory-less and shared by all agents, archetypal of collective behavior under simple rules (Hamann, 2018), and of interest to compute-constrained agents, including e.g. nano-particles or small robots. Optionally, memory and history-dependence can be integrated into the state, see Appendix E. Agents act according to policy $\pi \in \Pi$ from a class $\Pi \subseteq \mathcal{P}(\mathcal{U})^{\mathcal{Y} \times \mathcal{T}}$ of policies, with spaces of probability measures $\mathcal{P}(\cdot)$, equipped with the 1-Wasserstein metric $W_1$ (Villani, 2009). Starting with initial distribution $\mu_0$, $x_0^i \sim \mu_0$, the **MFC-type Dec-POMDP** dynamics are

$$y_t^i \sim P^y(y_t^i \mid x_t^i, \mu_t^N), \quad u_t^i \sim \pi_t(u_t^i \mid y_t^i), \quad x_{t+1}^i \sim P(x_{t+1}^i \mid x_t^i, u_t^i, \mu_t^N) \tag{1}$$

for all $(i, t) \in [N] \times \mathcal{T}$, with transition kernels $P \colon \mathcal{X} \times \mathcal{U} \times \mathcal{P}(\mathcal{X}) \to \mathcal{P}(\mathcal{X})$, $P^y \colon \mathcal{X} \times \mathcal{P}(\mathcal{X}) \to \mathcal{P}(\mathcal{Y})$, objective $J^N(\pi) = \mathbb{E}[\sum_{t \in \mathcal{T}} \gamma^t r(\mu_t^N)]$ to maximize over $\pi \in \Pi$ under reward function $r \colon \mathcal{P}(\mathcal{X}) \to \mathbb{R}$, and discount factor $\gamma \in (0, 1)$. Results generalize to finite horizons, average per-agent rewards $r_{\mathrm{per}} \colon \mathcal{X} \to \mathbb{R}$, $r(\mu_t^N) = \int r_{\mathrm{per}} \mathrm{d}\mu_t^N$, and joint state-observation-action MFs via enlarged state space.

Since general Dec-POMDPs are hard (Bernstein et al., 2002), our model establishes a tractable special case of high generality. Standard MFC already covers a broad range of applications, e.g. see surveys for finance (Carmona, 2020) and engineering (Djehiche et al., 2017) applications, which can now be handled under partial information. In addition, many classical, inherently partially observable models are covered by MFC-type Dec-POMDPs, such as the Kuramoto or Vicsek models in Section 4, where many-agent convergence is known as propagation of chaos (Chaintron & Diez, 2022).

## 2.2 LIMITING MFC SYSTEM

In order to achieve tractability for large multi-agent systems, the first step is to take the infinite-agent limit. By a law of large numbers (LLN), this allows us to describe large systems only by the MF $\mu_t$. Consider a representative agent as in (1) with states $x_0 \sim \mu_0$, $x_{t+1} \sim P(x_{t+1} \mid x_t, u_t, \mu_t)$, observations $y_t \sim P^y(y_t \mid x_t, \mu_t)$ and actions $u_t \sim \pi_t(u_t \mid y_t)$. Then, its state probability law replaces the empirical state distribution, informally $\mu_t = \mathcal{L}(x_t) \equiv \lim_{N \to \infty} \mu_t^N$. Looking only at the MF, we hence obtain the decentralized partially observable MFC (**Dec-POMFC**) system

$$\mu_{t+1} = \mathcal{L}(x_{t+1}) = T(\mu_t, \pi_t) := \iiint P(x, u, \mu_t) \pi_t(\mathrm{d}u \mid y) P^y(\mathrm{d}y \mid x, \mu_t) \mu_t(\mathrm{d}x) \tag{2}$$

by deterministic transitions $T \colon \mathcal{P}(\mathcal{X}) \times \mathcal{P}(\mathcal{U})^{\mathcal{Y}} \to \mathcal{P}(\mathcal{X})$ and objective $J(\pi) = \sum_{t=0}^{\infty} \gamma^t r(\mu_t)$.

**Approximation guarantees.** Under mild continuity assumptions, the Dec-POMFC model in (2) constitutes a good approximation of large-scale MFC-type Dec-POMDP in (1) with many agents.

**Assumption 1a.** *The transitions $P$, $P^y$ and rewards $r$ are Lipschitz with constants $L_P$, $L_{P^y}$, $L_r$.*

**Assumption 1b.** *The class of policies $\Pi$ is the set of all $L_\Pi$-Lipschitz policies for some $L_\Pi > 0$, i.e. for all $t \in \mathcal{T}$ and $\pi \in \Pi$, we have that $\pi_t \colon \mathcal{Y} \to \mathcal{P}(\mathcal{U})$ is $L_\Pi$-Lipschitz. Alternatively, we may assume unrestricted policies if (i) observations only depend on an agent's state, and (ii) $|\mathcal{X}| < \infty$.*

Lipschitz continuity of the model is commonly assumed (Huang et al., 2006; Gu et al., 2021; Mondal et al., 2022), and in general at least (uniform) continuity is required: Consider a counterexample with uniform initial $\mu_0$ over states $A, B$. If dynamics, observations, or rewards jump between regimes at $\mu(A) = \mu(B) = 0.5$, the finite system will randomly experience all regimes, while limiting MFC experiences only the regime at $\mu(A) = \mu(B) = 0.5$. Meanwhile, Lipschitz policies are not only standard in MFC literature (Pasztor et al., 2021; Mondal et al., 2022) by neural networks (NNs) (Araujo et al., 2023), but also fulfilled for finite $\mathcal{Y}$ trivially without loss of generality ($L_\Pi \coloneqq \mathrm{diam}(\mathcal{U})$), and for continuous $\mathcal{Y}$ by kernel parametrizations in Section 3. We extend MFC approximation theorems (Gu et al., 2021; Mondal et al., 2022; Cui et al., 2023) to partial observations and compact spaces.

**Theorem 1.** *Fix an equicontinuous family of functions $\mathcal{F} \subseteq \mathbb{R}^{\mathcal{P}(\mathcal{X})}$. Under Assumptions 1a–1b, the MF converges in the sense of $\sup_{\pi \in \Pi} \sup_{f \in \mathcal{F}} \mathbb{E}\left[\left|f(\mu_t^N) - f(\mu_t)\right|\right] \to 0$ at all times $t \in \mathcal{T}$.*

The approximation rate is $\mathcal{O}(1/\sqrt{N})$ for finite state-actions, using equi-Lipschitz $\mathcal{F}$ (Appendix D). Hence, the easier Dec-POMFC simplifies otherwise hard Dec-POMDPs. Indeed, we later show that such optimal Lipschitz Dec-POMFC policies are guaranteed to exist via closedness of joint-measures under equi-Lipschitz kernels (Appendix K), see Propositions 1, 2 and Theorem 2 later.

**Corollary 1.** *Under Assumptions 1a–1b, any optimal Dec-POMFC policy $\pi \in \arg\max_{\pi' \in \Pi} J(\pi')$ is $\varepsilon$-optimal in the MFC-type Dec-POMDP, $J^N(\pi) \geq \sup_{\pi' \in \Pi} J^N(\pi') - \varepsilon$, with $\varepsilon \to 0$ as $N \to \infty$.*

## 2.3 REWRITING POLICIES WITH MEAN FIELD OBSERVATIONS

Now introducing the next system for reduction to an MDP, writing $\bar{\mu}$, $\bar{\pi}$ etc., let policies depend also on $\mu_t$, i.e. policies "observe" the mean field. While we could reason that agents might observe the MF or use filtering to estimate it (Åström, 1965), more importantly, the limiting MF is *deterministic*. Therefore, w.l.o.g. we obtain the decentralized mean field observable MFC (**Dec-MFC**) dynamics

$$\bar{\mu}_{t+1} = T(\bar{\mu}_t, \bar{\pi}_t(\bar{\mu}_t)) \coloneqq \iiint P(x, u, \mu_t)\bar{\pi}_t(\mathrm{d}u \mid y, \bar{\mu}_t)P^y(\mathrm{d}y \mid x, \bar{\mu}_t)\bar{\mu}_t(\mathrm{d}x), \qquad (3)$$

with shorthand $\bar{\pi}_t(\bar{\mu}_t) = \bar{\pi}_t(\cdot \mid \cdot, \bar{\mu}_t)$, initial $\bar{\mu}_0 = \mu_0$ and according objective $\bar{J}(\bar{\pi}) = \sum_{t=0}^{\infty} \gamma^t r(\bar{\mu}_t)$ to optimize over (now MF-dependent) policies $\bar{\pi} \in \bar{\Pi} \subseteq \mathcal{P}(\mathcal{U})^{\mathcal{Y} \times \mathcal{P}(\mathcal{X}) \times \mathcal{T}}$.

Deterministic open-loop control transforms optimal *Dec-MFC* policies $\bar{\pi} \in \arg\max_{\bar{\pi}' \in \bar{\Pi}} \bar{J}(\bar{\pi}')$ into optimal *Dec-POMFC* policies $\pi \in \arg\max_{\pi \in \Pi} J(\pi)$ with decentralized execution, and vice versa: For given $\bar{\pi}$, compute deterministic MFs $(\bar{\mu}_0, \bar{\mu}_1, \ldots)$ via (3) and let $\pi = \Phi(\bar{\pi})$ by $\pi_t(\mathrm{d}u \mid y) = \bar{\pi}(\mathrm{d}u \mid y, \bar{\mu}_t)$. Analogously, represent $\pi \in \Pi$ by $\bar{\pi} \in \bar{\Pi}$ with constant $\bar{\pi}_t(\nu) = \pi_t$ for all $\nu$.

**Proposition 1.** *For any $\bar{\pi} \in \bar{\Pi}$, define $(\bar{\mu}_0, \bar{\mu}_1, \ldots)$ as in (3). Then, for $\pi = \Phi(\bar{\pi}) \in \Pi$, we have $\bar{J}(\bar{\pi}) = J(\pi)$. Inversely, for any $\pi \in \Pi$, let $\bar{\pi}_t(\bar{\nu}) = \pi_t$ for all $\bar{\nu}$, then again $\bar{J}(\bar{\pi}) = J(\pi)$.*

**Corollary 2.** *Optimal Dec-MFC policies $\bar{\pi} \in \arg\max_{\bar{\pi}' \in \bar{\Pi}} \bar{J}(\bar{\pi}')$ yield optimal Dec-POMFC policies $\Phi(\bar{\pi})$, i.e. $J(\Phi(\bar{\pi})) = \sup_{\pi' \in \Pi} J(\pi')$.*

Knowing initial $\mu_0$ is often realistic, as deployment is commonly for well-defined problems of interest. Even then, knowing $\mu_0$ is not strictly necessary (Section 4). In contrast to standard deterministic open-loop control, (i) agents have stochastic dynamics and observations, and (ii) agents randomize actions instead of playing a trajectory, still leading to quasi-deterministic MFs by the LLN.

## 2.4 REDUCTION TO DEC-MFC MDP

Lastly, we reformulate as an MDP with more tractable theory and algorithms, writing $\hat{\mu}$, $\hat{\pi}$ etc. The recent MFC MDP (Pham & Wei, 2018; Carmona et al., 2019b; Gu et al., 2019) reformulates *fully observable* MFC as MDPs with higher-dimensional state-actions. Similarly, we reduce Dec-MFC to an MDP with joint state-observation-action distributions as its MDP actions. The **Dec-MFC MDP** has states $\hat{\mu}_t \in \mathcal{P}(\mathcal{X})$ and actions $h_t \in \mathcal{H}(\hat{\mu}_t) \subseteq \mathcal{P}(\mathcal{X} \times \mathcal{Y} \times \mathcal{U})$ in the set of joint $h_t = \hat{\mu}_t \otimes P^y(\hat{\mu}_t) \otimes \check{\pi}_t$

under any $L_\Pi$-Lipschitz policy $\check{\pi}_t \in \mathcal{P}(\mathcal{U})^{\mathcal{Y}}$. Here, $\nu \otimes K$ is the product measure of measure $\nu$ and kernel $K$, and $\nu K$ is the measure $\nu K = \int K(\cdot \mid x)\nu(\mathrm{d}x)$. For $\check{\pi}_t \in \mathcal{P}(\mathcal{U})^{\mathcal{Y}}$, $\mu_{xy} \in \mathcal{P}(\mathcal{X} \times \mathcal{Y})$, we write $\mu_{xy} \otimes \check{\pi}_t$ by letting $\check{\pi}_t$ constant on $\mathcal{X}$. In other words, the desired joint $h_t$ results from all agents replacing the previous system's policy $\bar{\pi}_t$ by lower-level policy $\check{\pi}_t$, which may be reobtained from $h_t$ (Appendix K, disintegration (Kallenberg, 2021)). Equivalently, identify $\mathcal{H}(\mu)$ with $\mu$ and classes of $\check{\pi}_t$ yielding the same joint, and in practice we parametrize $\check{\pi}_t$. Thus, we obtain the MDP dynamics

$$h_t \sim \hat{\pi}(\hat{\mu}_t), \quad \hat{\mu}_{t+1} = \hat{T}(\hat{\mu}_t, h_t) := \iiint P(x, u, \hat{\mu}_t)h_t(\mathrm{d}x, \mathrm{d}y, \mathrm{d}u) \tag{4}$$

for Dec-MFC MDP policy $\hat{\pi} \in \hat{\Pi}$ and objective $\hat{J}(\hat{\pi}) = \mathbb{E}\left[\sum_{t=0}^{\infty} \gamma^t r(\hat{\mu}_t)\right]$. The Dec-MFC MDP policy $\hat{\pi}$ is "upper-level", as we sample $h_t$ from $\hat{\pi}$, to apply the lower-level policy $\check{\pi}_t[h_t]$ to all agents.

**Guidance by mean field dependence.** Intuitively, the MF *guides* policy search in potentially hard, decentralized problems, and reduces to a single-agent MDP where we make some existing theory compatible. First, we formulate a dynamic programming principle (DPP), i.e. exact solutions by Bellman's equation for the value function $V(\mu) = \sup_{h \in \mathcal{H}(\mu)} r(\mu) + \gamma V(\hat{T}(\mu, h))$ (Hernández-Lerma & Lasserre, 2012). Here, a central theoretical novelty is closedness of joint measures under equi-Lipschitz policies (Appendix K). Concomitantly, we obtain optimality of stationary deterministic $\hat{\pi}$. For technical reasons, only here we assume Hilbertian $\mathcal{Y}$ (e.g. finite or Euclidean) and finite $\mathcal{U}$.

**Assumption 2.** *The observations $\mathcal{Y}$ are a metric subspace of a Hilbert space. Actions $\mathcal{U}$ are finite.*

**Theorem 2.** *Under Assumptions 1a–1b and 2, there exists an optimal stationary, deterministic policy $\hat{\pi}$ for the Dec-MFC MDP, with $\hat{\pi}(\mu) \in \arg\max_{h \in \mathcal{H}(\mu)} r(\mu) + \gamma V(\hat{T}(\mu, h))$.*

**Decentralized execution.** Importantly, guidance by MF is only for training and not execution. An optimal upper-level policy $\hat{\pi} \in \arg\max_{\hat{\pi}' \in \hat{\Pi}} \hat{J}(\hat{\pi})$ is optimal also for the initial system, if it is deterministic, and an optimal one exists by Theorem 2. The lower-level policies $\bar{\pi}_t \equiv \check{\pi}_t$ are obtained by inserting the sequence of MFs $\hat{\mu}_0, \hat{\mu}_1, \ldots$ into $\hat{\pi}$, and remain non-stationary stochastic policies.

**Proposition 2.** *For deterministic $\hat{\pi} \in \hat{\Pi}$, let $\hat{\mu}_t$ as in (4) and $\bar{\pi} = \Psi(\hat{\pi})$ by $\bar{\pi}_t(\nu) = \check{\pi}_t$ for all $\nu$, then $\hat{J}(\hat{\pi}) = \bar{J}(\bar{\pi})$. Inversely, for $\bar{\pi} \in \bar{\Pi}$, let $\hat{\pi}_t(\nu) = \nu \otimes P^y(\nu) \otimes \bar{\pi}_t(\nu)$ for all $\nu$, then $\hat{J}(\hat{\pi}) = \bar{J}(\bar{\pi})$.*

Note that the determinism of the *upper-level policy* is strictly necessary: A simple counterexample is a problem where agents should choose to aggregate to one state. If the upper-level policy randomly chooses between moving all agents to either $A$ or $B$, then a corresponding random agent policy splits agents and fails to aggregate. At the same time, randomization of *agent actions* remains necessary for optimality, as the problem of equally spreading would require uniformly random agent actions.

**Complexity.** Tractability of multi-agent control heavily depends on information structure (Mahajan et al., 2012). General Dec-POMDPs have doubly-exponential complexity (NEXP, Bernstein et al. (2002)) and are harder than fully observable control (PSPACE, Papadimitriou & Tsitsiklis (1987)). In contrast, Dec-POMFC surprisingly imposes little additional complexity over standard MFC, as the MFC MDP remains deterministic in the absence of common noise correlating agents (Carmona et al., 2016). An analysis with common noise is possible, e.g. if observing the mean field, but out of scope.

## 3 DEC-POMFC POLICY GRADIENT METHODS

All that remains is to solve Dec-MFC MDPs. As we obtain continuous Dec-MFC MDP states and actions even for finite $\mathcal{X}, \mathcal{Y}, \mathcal{U}$, and infinite-dimensional ones for continuous $\mathcal{X}, \mathcal{Y}, \mathcal{U}$, a value-based approach can be hard. Our policy gradient (PG) approach allows finding simple policies for collective behavior, with emergence of global intelligent behavior described by rewards $r$, under arbitrary (Lipschitz) policies. For generality, we use NN upper-level and kernel lower-level policies. While lower-level (Lipschitz, (Araujo et al., 2023)) NNs policies could be considered akin to hypernetworks (Ha et al., 2016), the resulting distributions over NN parameters as MDP actions are too high-dimensional and failed in our experiments. We directly solve finite-agent MFC-type Dec-POMDPs by solving the Dec-MFC MDP in the background. Indeed, the **theoretical optimality** of Dec-MFC MDP solutions is guaranteed over Lipschitz policies in $\Pi$.

**Corollary 3.** *Under Assumptions 1a–1b, a deterministic Dec-MFC solution $\hat{\pi} \in \arg\max_{\hat{\pi}'} \hat{J}(\hat{\pi}')$ is $\epsilon$-optimal in the Dec-POMDP, $J^N(\Phi(\Psi(\hat{\pi}))) \geq \sup_{\pi' \in \Pi} J^N(\pi') - \epsilon$, with $\epsilon \to 0$ as $N \to \infty$.*

**Histogram vs. kernel parametrizations.** Except for linear-quadratic algorithms (Wang et al., 2021; Fu et al., 2019; Carmona et al., 2019a), the only approach to learning MFC in continuous spaces $\mathcal{X} \subseteq \mathbb{R}^n$, $n \in \mathbb{N}$ (and here $\mathcal{Y}$) is by partitioning and "discretizing" (Carmona et al., 2019b; Gu et al., 2021).[1] Unfortunately, partitions fail Lipschitzness and hence approximation guarantees, even in standard MFC. Instead, we use kernel representations for MFs $\mu_t^N$ and lower-level policies $\check{\pi}_t$.

We represent $\mathcal{P}(\mathcal{X})$-valued MDP states $\mu_t^N$ not by counting agents in each bin, but instead mollify around each center $x_b \in \mathcal{X}$ of $M_{\mathcal{X}}$ bins $b \in [M_{\mathcal{X}}]$ using kernels. The result is Lipschitz and approximates histograms arbitrarily well (Miculescu, 2000, Theorem 1). Hence, we obtain input logits $I_b = \int \kappa(x_b, \cdot) \mathrm{d}\mu_t^N = \frac{1}{N} \sum_{i \in [N]} \kappa(x_b, x_t^i)$ for some kernel $\kappa \colon \mathcal{X} \times \mathcal{X} \to \mathbb{R}$ and $b \in [M_{\mathcal{X}}]$. Output logits constitute mean and log-standard deviation of a diagonal Gaussian over parameter representations $\xi \in \Xi$ of $\check{\pi}_t$. We obtain Lipschitz $\check{\pi}_t$ by representing $\check{\pi}_t$ via $M_{\mathcal{Y}}$ points $y_b \in \mathcal{Y}$ such that $\check{\pi}_t(u \mid y) = \sum_{b \in [M_{\mathcal{Y}}]} \kappa(y_b, y) p_b(u) / \sum_{b \in [M_{\mathcal{Y}}]} \kappa(y_b, y)$. Here, we consider $L_\lambda$-Lipschitz maps $\lambda_b$ from parameters $\xi \in \Xi$ to distributions $p_b = \lambda_b(\xi) \in \mathcal{P}(\mathcal{U})$ with compact parameter space $\Xi$, and for kernels choose RBF kernels $\kappa(x, y) = \exp(-\|x - y\|^2 / (2\sigma^2))$ with some bandwidth $\sigma^2 > 0$.

**Proposition 3.** *Under RBF kernels $\kappa$, for any $\xi$ and Euclidean $\mathcal{Y}$, lower-level policies $\Lambda(\xi)(\cdot \mid y) :=$ $\sum_{b \in [M_{\mathcal{Y}}]} \kappa(y_b, y) \lambda_b(\xi) / \sum_{b \in [M_{\mathcal{Y}}]} \kappa(y_b, y)$ are $L_\Pi$-Lipschitz in $y$ as in Assumption 1b, whenever $\sigma^2 \exp^2\left(-\frac{1}{2\sigma^2} \mathrm{diam}(\mathcal{Y})^2\right) \geq \frac{1}{L_\Pi} \mathrm{diam}(\mathcal{Y}) \mathrm{diam}(\mathcal{U}) \max_{y \in \mathcal{Y}} \|y\|$, and such $\sigma^2 > 0$ always exists.*

Proposition 3 ensures Assumption 1b if needed. To achieve optimality by Corollary 3, deterministic policies commonly result from convergence of stochastic PGs, taking mean actions, or are guaranteed by deterministic PGs (Silver et al., 2014; Lillicrap et al., 2016). Beyond allowing for (i) Lipschitz guarantees, and (ii) finer control over agent actions, another advantage of kernels is (iii) the improved complexity over histograms. Even a histogram with only 2 bins per dimension requires $2^d$ bins in $d$-dimensional spaces, while kernel representations may place e.g. 2 points per dimension, improving upon the otherwise necessarily exponential complexity, see also Appendix A for empirical support.

**Direct multi-agent reinforcement learning algorithm.** Applying RL directly to the Dec-MFC MDP would be satisfactory only under known MFC models. Importantly, (i) we do not always have access to the model, and (ii) even if we do, parametrizing MFs in arbitrary compact $\mathcal{X}$ is hard. Instead, it is more practical and tractable to train on a finite system. Our direct MARL approach hence trains on a finite $N$-agent MFC-type Dec-POMDP of interest, in a model-free manner. In order to exploit the underlying MDP, our algorithm assumes *during training* that (i) the MF is observed, and (ii) agents can correlate actions (e.g. centrally, or sharing seeds). Therefore, the finite system (1) is adjusted for training by correlating agent actions on a single centrally sampled lower-level policy $\check{\pi}_t$. Now write $\hat{\pi}^\theta(\xi_t \mid \tilde{\mu}_t^N)$ as density over parameters $\xi_t \in \Xi$ under a base measure (discrete, Lebesgue). Substituting $\xi_t$ as actions parametrizing $h_t$ in the MDP (4), e.g. by using RBF kernels, yields the **centralized training** system as seen in Figure 1 for stationary policy $\hat{\pi}^\theta$ parametrized by $\theta$,

$$\check{\pi}_t = \Lambda(\tilde{\xi}_t), \quad \tilde{\xi}_t \sim \hat{\pi}^\theta(\tilde{\mu}_t^N),$$
$$\tilde{y}_t^i \sim P^y(\tilde{y}_t^i \mid \tilde{x}_t^i, \tilde{\mu}_t^N), \quad \tilde{u}_t^i \sim \check{\pi}_t(\tilde{u}_t^i \mid \tilde{y}_t^i), \quad \tilde{x}_{t+1}^i \sim P(\tilde{x}_{t+1}^i \mid \tilde{x}_t^i, \tilde{u}_t^i, \tilde{\mu}_t^N), \quad \forall i \in [N]. \tag{5}$$

**Policy gradient approximation.** Since we train on a finite system, it is not immediately clear whether centralized training really yields the PG for the underlying Dec-MFC MDP, also in existing literature for learning MFC. We will show this practically relevant fact up to an approximation. The general PG for stationary $\hat{\pi}^\theta$ (Sutton et al., 1999; Peters & Schaal, 2008) is $\nabla_\theta J(\hat{\pi}^\theta) = (1 - \gamma)^{-1} \mathbb{E}_{\mu \sim d_{\hat{\pi}^\theta}, \xi \sim \hat{\pi}^\theta(\mu)} \left[ Q^\theta(\mu, \xi) \nabla_\theta \log \hat{\pi}^\theta(\xi \mid \mu) \right]$ with $Q^\theta(\hat{\mu}, \xi) = \mathbb{E}[\sum_{t=0}^\infty \gamma^t r(\hat{\mu}_t) \mid \hat{\mu}_0 = \mu, \xi_0 = \xi]$ under parametrized actions $\xi_t$ in (4), and using sums $d_{\hat{\pi}^\theta} = (1 - \gamma) \sum_{t \in \mathcal{T}} \gamma^t \mathcal{L}_{\hat{\pi}^\theta}(\hat{\mu}_t)$ of laws of $\hat{\mu}_t$ under $\hat{\pi}^\theta$. Our approximation motivates MFC for MARL by showing that the *underlying background Dec-MFC MDP* is approximately solved under Lipschitz parametrizations, e.g. we normalize parameters $\xi$ to finite action probabilities, or use bounded diagonal Gaussian parameters.

**Assumption 3.** *The policy $\hat{\pi}^\theta(\xi \mid \mu)$ and its log-gradient $\nabla_\theta \log \hat{\pi}^\theta(\xi \mid \mu)$ are $L_{\hat{\Pi}}$, $L_{\nabla \hat{\Pi}}$-Lipschitz in $\mu$ and $\xi$ (or alternatively in $\mu$ for any $\xi$, and uniformly bounded). The parameter-to-distribution map is $\Lambda(\xi)(\cdot \mid y) := \sum_b \kappa(y_b, y) \lambda_b(\xi)(\cdot) / \sum_b \kappa(y_b, y)$, with kernels $\kappa$ and $L_\lambda$-Lipschitz $\lambda_b \colon \Xi \to \mathcal{P}(\mathcal{U})$.*

---

[1]Existing Q-Learning with kernel regression (Gu et al., 2021) is for *finite* states $\mathcal{X}$ with kernels on $\mathcal{P}(\mathcal{X})$, and learns on the MFC MDP. We allow *continuous* $\mathcal{Y}$ by kernels on $\mathcal{Y}$ itself, and learn on the finite-agent system.

---

**Algorithm 1** Dec-POMFPPO (during centralized training)

---

1: **for** iteration $n = 1, 2, \ldots$ **do**
2:     **for** time $t = 0, \ldots, B_{\text{len}} - 1$ **do**
3:         Sample central Dec-MFC MDP action $\check{\pi}_t = \Lambda(\xi_t)$, $\xi_t \sim \hat{\pi}^\theta(\tilde{\mu}_t^N)$.
4:         **for** agent $i = 1, \ldots, N$ **do**
5:             Sample per-agent action $\tilde{u}_t^i \sim \check{\pi}_t(\tilde{u}_t^i \mid \tilde{y}_t^i)$ for observation $\tilde{y}_t^i$.
6:         Perform actions, observe reward $r(\tilde{\mu}_t^N)$, next MF $\tilde{\mu}_{t+1}^N$, termination flag $d_{t+1} \in \{0, 1\}$.
7:     **for** updates $i = 1, \ldots, N_{\text{PPO}}$ **do**
8:         Sample mini-batch $b$, $|b| = b_{\text{len}}$ from data $B := ((\tilde{\mu}_t^N, \xi_t, r_t^N, d_{t+1}, \tilde{\mu}_{t+1}^N))_{t \geq 0}$.
9:         Update policy $\hat{\pi}^\theta$ via PPO loss $\nabla_\theta L_\theta$ on $b$, using GAE (Schulman et al., 2016).
10:         Update critic $V^{\theta'}$ via critic $L_2$-loss $\nabla_{\theta'} L_{\theta'}$ on $b$.

---

**Theorem 3.** *Centralized training on system* (5) *approximates the true gradient of the underlying Dec-MFC MDP, i.e. under RBF kernels $\kappa$ as in Proposition 3, Assumptions 1a–1b and 3, as $N \to \infty$,*

$$\left\| (1 - \gamma)^{-1} \, \mathbb{E}_{\mu \sim d_{\hat{\pi}^\theta}^N, \xi \sim \hat{\pi}^\theta(\mu)} \left[ \tilde{Q}^\theta(\mu, \xi) \nabla_\theta \log \hat{\pi}^\theta(\xi \mid \mu) \right] - \nabla_\theta J(\hat{\pi}^\theta) \right\| \to 0$$

*with $d_{\hat{\pi}^\theta}^N = (1 - \gamma) \sum_{t \in \mathcal{T}} \gamma^t \mathcal{L}_{\hat{\pi}^\theta}(\tilde{\mu}_t^N)$ and $\tilde{Q}^\theta(\mu, \xi) = \mathbb{E} \left[ \sum_{t=0}^\infty \gamma^t r(\tilde{\mu}_t^N) \mid \mu_0 = \mu, \xi_0 = \xi \right]$.*

The value function $\tilde{Q}^\theta$ in the finite system is then substituted in actor-critic manner by on-policy and critic estimates. The Lipschitz conditions of $\hat{\pi}^\theta$ in Assumption 3 are fulfilled by Lipschitz NNs (Pasztor et al., 2021; Mondal et al., 2022; Araujo et al., 2023) and our parametrizations. The approximation is novel, building a foundation for MARL via MFC directly on a finite MARL problem. Our results also apply to fully observable MFC by $y_t = x_t$. Though gradient estimates allow convergence guarantees in finite MDPs (e.g. Qu et al. (2020, Theorem 5)), Dec-MFC MDP state-actions are always non-finite. In practice, we use empirically more efficient proximal policy optimization (PPO, Schulman et al. (2017); Yu et al. (2022)) to obtain the *decentralized partially observable mean field PPO* algorithm (Dec-POMFPPO, Algorithm 1).

By Theorem 3, we may learn directly on the MFC-type Dec-POMDP system (1). During training, the algorithm (i) assumes to observe the MF, and (ii) samples only one centralized $h_t$. Knowledge of the MF during training aligns our framework with the popular *centralized training, decentralized execution* (CTDE) paradigm. During execution, decentralized policies suffice for near-optimality by Corollary 3 without agents knowing the MF or coordinating centrally. Decentralized training can also be achieved, if the MF is observable and all agents use the same seed to correlate their actions.

## 4 EVALUATION

In this section, we empirically evaluate our algorithm, comparing against *independent* and *multi-agent PPO* (IPPO, MAPPO) with state-of-the-art performance (Yu et al., 2022; Papoudakis et al., 2021). For comparison, we share hyperparameters and architectures between algorithms, see Appendices A–C.

**Problems.** In the **Aggregation** problem we consider a typical continuous single integrator model, commonly used in the study of swarm robotics (Soysal & Sahin, 2005; Bahgeçi & Sahin, 2005). Agents observe their own position noisily and should aggregate. The classical **Kuramoto** model is used to study synchronization of coupled oscillators, finding application not only in physics, including quantum computation and laser arrays (Acebrón et al., 2005), but also in diverse biological systems, such as neuroscience and pattern formation in self-organizing systems (Breakspear et al., 2010; Kruk et al., 2020). Here, via partial observability, we consider a version where each oscillator can see the distribution of relative phases of its neighbors. Finally, we implement the Kuramoto model on a random geometric graph (e.g. (Diaz-Guilera et al., 2009)) via omitting movement in its independent generalization, the **Vicsek** model (Vicsek et al., 1995; Vicsek & Zafeiris, 2012). Agents $j$ have two-dimensional position $p_t^j$ and current headings $\phi_t^j$, to be controlled by their actions. The key metric of interest for both Kuramoto and Vicsek is polarization via the *polar order parameter* $R = |\frac{1}{N} \sum_j \exp(i\phi_t^j)|$. Here, $R$ ranges from $0$ – fully unsynchronized – to $1$ – perfect alignment of agents. Experimentally, we consider various environments, such as the torus, Möbius strip, projective plane and Klein bottle. Importantly, agents only observe relative headings of others.

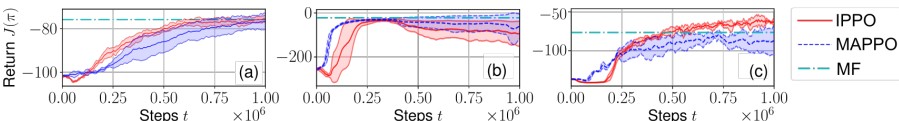

Figure 3: Dec-POMFPPO training curves (episode return) with shaded standard deviation over 3 seeds for $N = 200$ in (a) Aggregation; Vicsek on a (b): torus; (c): Möbius strip; (d): projective plane; (e): Klein bottle; and (f) Kuramoto on a torus.

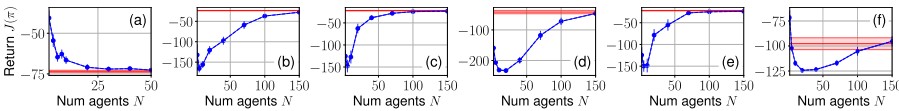

Figure 4: Training curves (episode return) with shaded standard deviation over 3 seeds and $N = 200$, in (a) Aggregation (box), (b) Vicsek (torus), (c) Kuramoto (torus). For comparison, we also plot the best return averaged over 3 seeds for Dec-POMFPPO in Figure 3 (MF).

**Training results.** In Figure 3 it is evident that the training process of MFC for many agents is relatively stable by guidance via MF and reduction to single-agent RL. In Appendix A, we also see similar results with significantly fewer agents and comparable to the results obtained with a larger number of agents. This observation highlights that the training procedure yields satisfactory outcomes, even in scenarios where the mean field approximation may not yet be perfectly exact. These findings underscore the generality of the proposed framework and its ability to adapt across different regimes. On the same note, we see by comparison with Figure 4, that our method is usually on par with state-of-the-art IPPO and MAPPO for many agents, e.g. here $N = 200$.

**Verification of theory.** In Figure 5, as the number of agents rises, the performance quickly tends to its limit, i.e. the objective converges, supporting Theorem 1 and Corollary 1, as well as applicability to arbitrarily many agents. Analogously, conducting open-loop experiments on our closed-loop trained system in Figure 6 demonstrates the robust generality of learned collective behavior with respect to the randomly sampled initial agent states, supporting Theorem 3 and Corollary 2.

**Qualitative analysis.** In the Vicsek model, as seen exemplarily in Figure 6 and Appendix A, the algorithm learns to align in various topological spaces. In all considered topologies, the polar order parameter surpasses $0.9$, with the torus system even reaching a value close to $0.99$. As for the angles at different iterations of the training process, as depicted in Figure 7, the algorithm gradually learns to form a concentrated cluster of angles. Note that the cluster center angle is not fixed, but rather changes over time. This behavior can not be observed in the classical Vicsek model, though extensions using more sophisticated equations of motion for angles have reported similar results (Kruk et al., 2020). For more details and further experiments or visualizations, we refer the reader to Appendices A–C. Figure 7 and additional figures, with similar results for other topologies in Appendix A, e.g. Figures 18–22, illustrate the qualitative behavior observed across the different manifolds. Agents on the continuous torus demonstrate no preference for a specific direction across consecutive training runs. Conversely, agents trained on other manifolds exhibit a tendency to avoid the direction that leads to an angle flip when crossing the corresponding boundary. Especially for the projective plane topology, the agents tend to aggregate more while aligning, even without adding another reward for aggregation. For Aggregation in Figure 8, we also find successful aggregation

Figure 5: The performance of the best of 3 Dec-POMFPPO policies transferred to $N$-agent systems (in blue, error bars for $95\%$ confidence interval), averaged over 50 episodes, and compared against the performance in the training system (in red). Problems (a)-(f) and training are as in Figure 3.

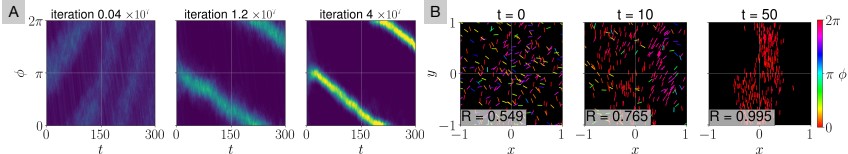

Figure 6: A, B: For the Vicsek (torus) problem with forward velocity control, the open-loop behavior (B) shows little difference in performance of agents (rods, color indicating heading) over the closed-loop behavior (A). C: Visualization of agents (triangles) under the Vicsek model on the torus.

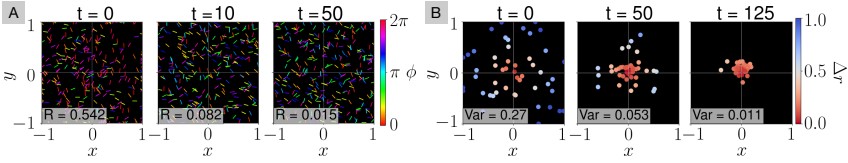

Figure 7: A: Agent angle alignment in the Vicsek model on the torus, plotted as density over time; B: Alignment of agents in the Vicsek model on the projective plane, as in Figure 6.

of agents in the middle. In practice, one may define any objective of interest. For example, we can achieve misalignment in Figure 8, resulting in polar order parameters on the order of magnitude of $10^{-2}$, and showing the generality of the framework.

**Additional experiments.** Some other experiments are discussed in Appendix A, including the generalization of our learned policies to different starting conditions, a comparison of the Vicsek model trained or transferred to different numbers of agents, additional interpretative visualizations, similar success for the Kuramoto model, and a favorable comparison between RBFs and histograms for higher dimensions, showing the generality of the framework and supporting our claims.

## 5 CONCLUSION AND DISCUSSION

Our framework provides a novel methodology for engineering artificial collective behavior in a rigorous and tractable manner, whereas existing scalable learning frameworks often focus on competitive or fully observable models (Guo et al., 2023; Zheng et al., 2018). We hope our work opens up new applications of partially-observable swarm systems. Our method could be of interest due to (i) its theoretical optimality guarantees while covering a large class of problems, and (ii) its surprising simplicity in rigorously reducing complex Dec-POMDPs to MDPs, with same complexity as MDPs from fully observable MFC, thus allowing analysis of Dec-POMDPs via a tractable MDP.

The current theory remains limited to non-stochastic MFs, which in the future could be analyzed for stochastic MFs via common noise (Perrin et al., 2020; Cui et al., 2023; Dayanikli et al., 2023). Further, sample efficiency could be analyzed (Huang et al., 2023), and parametrizations for history-dependent policies using more general NNs could be considered, e.g. via hypernetworks (Ha et al., 2016; Li et al., 2023). Lastly, extending the framework to consider additional practical constraints and sparser interactions, such as physical collisions or via graphical decompositions, may be fruitful.

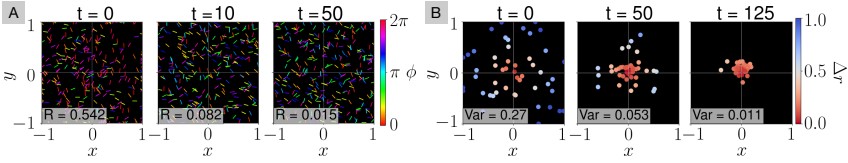

Figure 8: A: Qualitative behavior for misalignment of agents in the Vicsek (torus) problem. B: The two-dimensional Aggregation problem, with agent distances to mean as colors.

ACKNOWLEDGMENTS

This work has been co-funded by the LOEWE initiative (Hesse, Germany) within the emergenCITY center and the FlowForLife project, and the Hessian Ministry of Science and the Arts (HMWK) within the projects "The Third Wave of Artificial Intelligence - 3AI" and hessian.AI. The authors acknowledge the Lichtenberg high performance computing cluster of the TU Darmstadt for providing computational facilities for the calculations of this research.

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

## A  ADDITIONAL EXPERIMENTS

In this section, we give additional details on experiments. The mathematical description of problems can be found in Appendix C.

We use the manifolds as depicted in Figure 9 and as described in the following. Here, we visualize the qualitative results as in the main text for the remaining topologies. Due to technical limitations, all agents are drawn, including the ones behind a surface. To indicate where an agent belongs, we colorize the inside of the agent with the color of its corresponding surface.

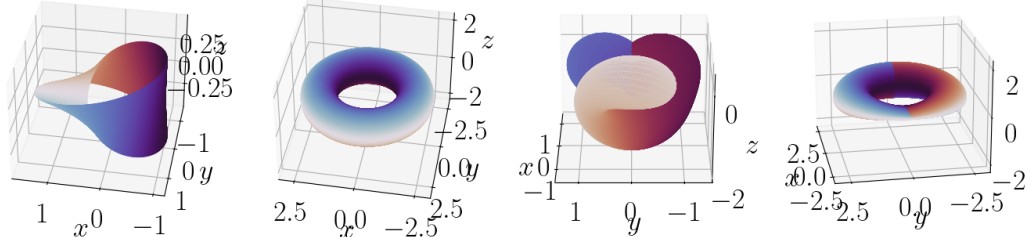

Figure 9: Two-dimensional manifolds visualized in three-dimensional space. In order from left to right: Möbius strip, torus, projective plane (Boy's surface), and Klein bottle (pinched torus).

**Torus manifold.**   The (flat) torus manifold is obtained from the square $[-1, 1]^2$ by identifying $(x, -1) \sim (x, 1)$ and $(-1, y) \sim (1, y)$ for all $x, y \in [-1, 1]$. For the metric, we use the toroidal distance inherited from the Euclidean distance $d_2$, which can be computed as

$$d(x, y) = \min_{t_1, t_2 \in \{-1, 0, 1\}} d_2(x, y + 2t_1 e_1 + 2t_2 e_2)$$

where $e_1, e_2$ denote unit vectors. In Figure 9, we visualize the torus by mapping each point $(x, y) \in [-1, 1]^2$ to a point $(X, Y, Z)$ in 3D space, given by

$$
\begin{aligned}
X &= (2 + 0.75 \cos(\pi(x + 1))) \cos(\pi(y + 1)), \\
Y &= (2 + 0.75 \cos(\pi(x + 1))) \sin(\pi(y + 1)), \\
Z &= 0.75 \sin(\pi(x + 1)).
\end{aligned}
$$

The results have been described in the main text in Figure 7. Here, also note that the torus – by periodicity and periodic boundary conditions – can essentially be understood as the case of an *infinite plane*, consisting of infinitely many copies of the square $[-1, 1]^2$ laid next to each other.

**Möbius strip.**   The Möbius strip is obtained from the square $[-1, 1]^2$ by instead only identifying $(-1, -y) \sim (1, y)$ for all $y \in [-1, 1]$, i.e. only the top and bottom side of the square, where directions are flipped. We then use the inherited distance

$$d(x, y) = \min_{t_2 \in \{-1, 0, 1\}} d_2(x, (1 - 2 \cdot \mathbf{1}_{t_2 \neq 0}, 1)^T \odot y + 2t_2 e_2)$$

where $\odot$ denotes the elementwise (Hadamard) product.

We visualize the Möbius strip in Figure 9 by mapping each point $(x, y) \in [-1, 1]^2$ to

$$
\begin{aligned}
X &= \left(1 + \frac{x}{2} \cos\left(\frac{\pi}{2}(y + 1)\right)\right) \cos(\pi(y + 1)), \\
Y &= \left(1 + \frac{x}{2} \cos\left(\frac{\pi}{2}(y + 1)\right)\right) \sin(\pi(y + 1)), \\
Z &= \frac{x}{2} \sin\left(\frac{\pi}{2}(y + 1)\right).
\end{aligned}
$$

As we can see in Figure 10, the behavior of agents is learned as expected: Agents learn to align along one direction on the Möbius strip.

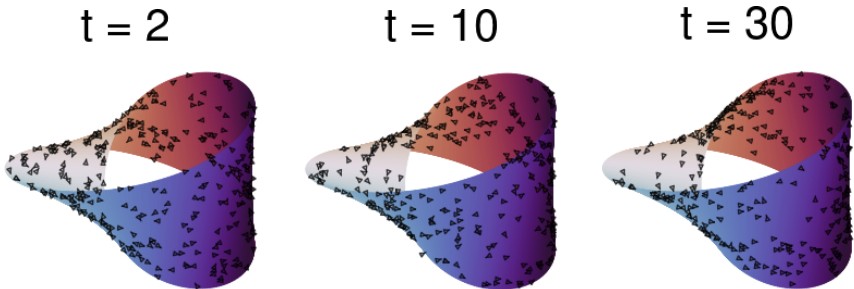

Figure 10: Qualitative visualization of Vicsek behavior on the Möbius strip manifold for uniform initialization. The 300 agents (triangles) align into one direction on the Möbius strip.

**Projective plane.** Analogously, the projective plane is obtained by identifying and flipping both sides of the square $[-1,1]^2$, i.e. $(-x,-1) \sim (x,1)$ and $(-1,-y) \sim (1,y)$ for all $x,y \in [-1,1]$. We use the inherited distance

$$d(x,y) = \min_{t_1,t_2 \in \{-1,0,1\}} d_2(x, (1 - 2 \cdot \mathbf{1}_{t_2 \neq 0}, 1 - 2 \cdot \mathbf{1}_{t_1 \neq 0})^T \odot y + 2t_1 e_1 + 2t_2 e_2)$$

and though an accurate visualization in less than four dimensions is difficult, we visualize in Figure 11 by mapping each point $(x,y) \in [-1,1]^2$ to a point $(X,Y,Z)$ on the so-called Boy's surface, with

$$z = \frac{x+1}{2} \exp\left(i\pi(y+1)\right),$$

$$g_1 = -\frac{3}{2} \operatorname{Im}\left(\frac{z(1-z^4)}{z^6 + \sqrt{5}z^3 - 1}\right),$$

$$g_2 = -\frac{3}{2} \operatorname{Re}\left(\frac{z(1-z^4)}{z^6 + \sqrt{5}z^3 - 1}\right),$$

$$g_3 = \operatorname{Im}\left(\frac{1+z^6}{z^6 + \sqrt{5}z^3 - 1}\right) - \frac{1}{2},$$

$$X = \frac{g_1}{g_1^2 + g_2^2 + g_3^2},$$

$$Y = \frac{g_2}{g_1^2 + g_2^2 + g_3^2},$$

$$Z = \frac{g_2}{g_1^2 + g_2^2 + g_3^2}.$$

As we can see in Figure 11, under the inherited metric and radial parametrization, agents tend to gather at the bottom of the surface.

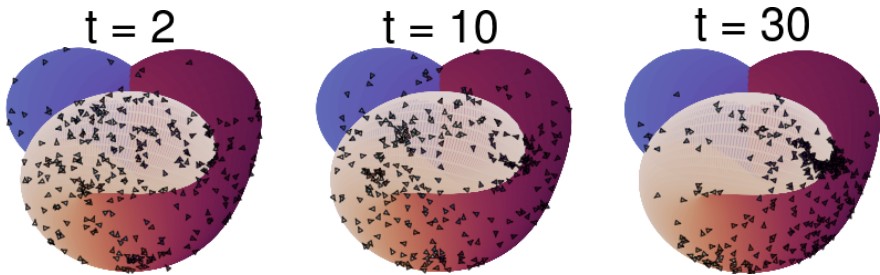

Figure 11: Qualitative visualization of Vicsek behavior on the projective plane manifold for uniform initialization. The 300 agents (triangles) align over time by gathering at the bottom right.

**Klein bottle.** Similarly, the Klein bottle is obtained by identifying both sides of the square $[-1,1]^2$ and flipping one side, i.e. $(x,-1) \sim (x,1)$ and $(-1,-y) \sim (1,y)$ for all $x,y \in [-1,1]$. We use the

inherited distance

$$d(x,y) = \min_{t_1,t_2 \in \{-1,0,1\}} d_2(x, (1 - 2 \cdot \mathbf{1}_{t_2 \neq 0}, 1)^T \odot y + 2t_1 e_1 + 2t_2 e_2)$$

and visualize in Figure 12 by the pinched torus, i.e. mapping each $(x, y) \in [-1, 1]^2$ to a point $(X, Y, Z)$ with

$$X = (2 + 0.75 \cos(\pi(x+1))) \cos(\pi(y+1)),$$
$$Y = (2 + 0.75 \cos(\pi(x+1))) \sin(\pi(y+1)),$$
$$Z = 0.75 \sin(\pi(x+1)) \cos\left(\frac{\pi}{2}(y+1)\right).$$

As we can see in Figure 12, agents may align by aggregating on the inner and outer ring, such that they may avoid switching sides at the pinch.

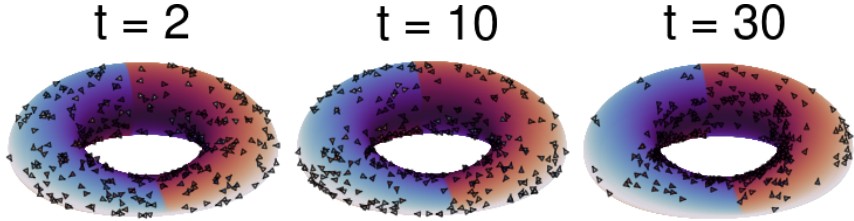

Figure 12: Qualitative visualization of behavior on Klein bottle topology for uniform initialization: The visualization in 3D is limited. Here, we use a pinched torus visualization that inverts itself at the flat pinch (i.e. there is no "connection" between blue and red surfaces at the bottom). Over time, 300 agents (triangles) sometimes align by aggregating on the inner, avoiding side switches at the pinch.

**Box.** Lastly, the box manifold is the square $[-1, 1]^2$ equipped with the standard Euclidean topology, i.e. distances between two points $x, y \in [-1, 1]^2$ are given by

$$d(x, y) = \sqrt{(x_1 - y_1)^2 + (x_2 - y_2)^2},$$

while the sides of the square are not connected to anything else. We use the box manifold for the following experiments in Aggregation, and mention it here for sake of completeness.

**Ablation on number of agents.** As seen in Figures 13 and 14, we can successfully train on various numbers of agents, despite the inaccuracy of the mean field approximation for fewer agents as inferred from Figure 5. This indicates that our algorithm is general and – at least in the considered problems – scales to arbitrary numbers of agents.

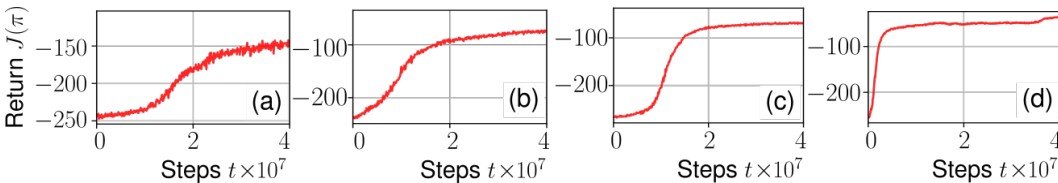

Figure 13: Training curves for Vicsek (torus), using RBF-based or discretization-based solutions. (a): RBF, $N = 25$; (b): Discretization, $N = 25$; (c): RBF, $N = 50$; (d) : Discretization, $N = 50$.

**Qualitative results for Kuramoto.** The Kuramoto model, see Figure 15, demonstrates instability during training and subsequent lower-grade qualitative behavior compared to the Vicsek model. This disparity persists even when considering more intricate topologies, despite being a specialization of the Vicsek model. One explanation is that the added movement makes it easier to align agents over time. Another general explanation could be that, despite initially distributing agents uniformly across the region of interest, the learned policy causes the agents to aggregate into a few or even a

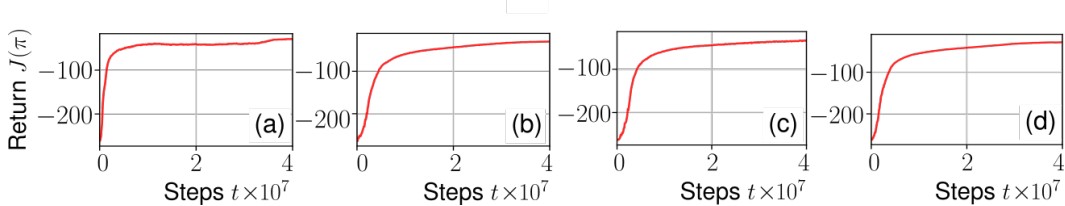

Figure 14: Training curves for Vicsek (torus), using RBF-based or discretization-based solutions. (a): RBF, $N = 100$; (b): Discretization, $N = 100$; (c): RBF, $N = 150$; (d) : Discretization, $N = 150$.

single cluster (though we do not observe such behavior in Figure 15). The closer particles are to each other, the greater the likelihood that they perceive a similar or identical mean field, prompting alignment only in local clusters. A similar behavior is observed in the classical Vicsek model, where agents tend to move in the same direction after interaction. Consequently, they remain within each other's interaction region and have the potential to form compounds provided there are no major disturbances. These can come from either other particles or excessively high levels of noise (Barberis, 2017). Although agents are able to align, the desired alignment remains to be improved, either via more parameter tuning or improved algorithms.

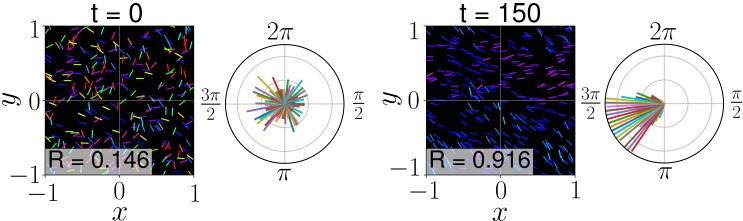

Figure 15: Qualitative behavior of the learned behavior in the Kuramoto model with histogram over angles, where in contrast to Vicsek, agents remain fixed in their initial position.

**Effect of kernel method.** While for low dimensions, the effect of kernel methods is not as pronounced and mostly ensure theoretical guarantees, in Figures 16 and 17 we can see that training via our RBF-based methods outperforms discretization-based methods for dimensions higher than 3 as compared to a simple gridding of the probability simplex with associated histogram representation of the mean field. Here, for the RBF method in Aggregation, we place 5 equidistant points $y_b$ on the axis of each dimension. This is also the reason for why the discretization-based approach is better for low dimensions $d = 2$ or $d = 3$, as more points will have more control over actions of agents, and can therefore achieve better results, in exchange for tractability in high dimensions. This shows the advantage of RBF-based methods in more complex, high-dimensional problems. While the RBF-based method continues to learn even for higher dimensions up to $d = 5$, the discretization-based solution eventually stops learning due to very large action spaces leading to increased noise on the gradient. The advantage is not just in terms of sample complexity, but also in terms of wall clock time, as the computation over exponentially many bins also takes more CPU time as shown in Table 1.

Table 1: Wall clock time for one training step averaged over 500 iterations in $d$-dimensional Aggregation, for 50 agents.

| Dimensionality $d$ | RBF MFC [s] | Discretization MFC [s] | MARL (IPPO) [s] |
|---|---|---|---|
| 2 | 5.64 | 5.69 | 146.58 |
| 3 | 6.16 | 7.96 | 147.03 |
| 4 | 6.97 | 17.26 | 147.29 |
| 5 | 8.31 | 76.33 | 146.91 |

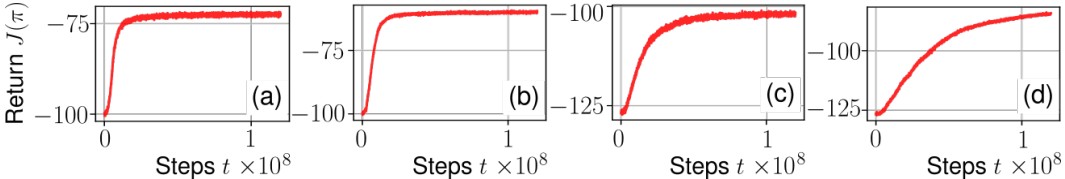

Figure 16: Training curves for $d$-dimensional Aggregation, using RBF-based vs. discretization-based solutions. (a): RBF, $d = 2$; (b): Discretization, $d = 2$; (c): RBF, $d = 3$; (d) : Discretization, $d = 3$.

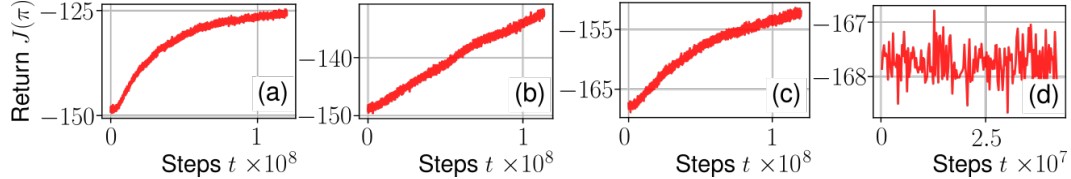

Figure 17: Training curves for $d$-dimensional Aggregation, using RBF-based vs. discretization-based solutions. (a): RBF, $d = 4$; (b): Discretization, $d = 4$; (c): RBF, $d = 5$; (d) : Discretization, $d = 5$.

**Ablations on time dependency and starting conditions.** As discussed in the main text, we also verify the effect of using a non-time-dependent open-loop sequence of lower-level policies, and also an ablation over different starting conditions. In particular, for starting conditions, to begin we will consider the **uniform** initialization as well as the **beta-1**, **beta-2** and **beta-3** initializations with a beta distribution over each dimension of the states, using $\alpha = \beta = 0.5$, $\alpha = \beta = 0.25$ and $\alpha = \beta = 0.75$ respectively.

As we can see in Figure 18, the behavior learned for the Vicsek problem on the torus with $N = 200$ agents allows for using the first lower-level policy $\check{\pi}_0$ at all times $t$ under the Gaussian initialization used in training to nonetheless achieve alignment. This validates the fact that a time-variant open-loop sequence of lower-level policies is not always needed, and the results even hold for slightly different initial conditions from the ones used in training.

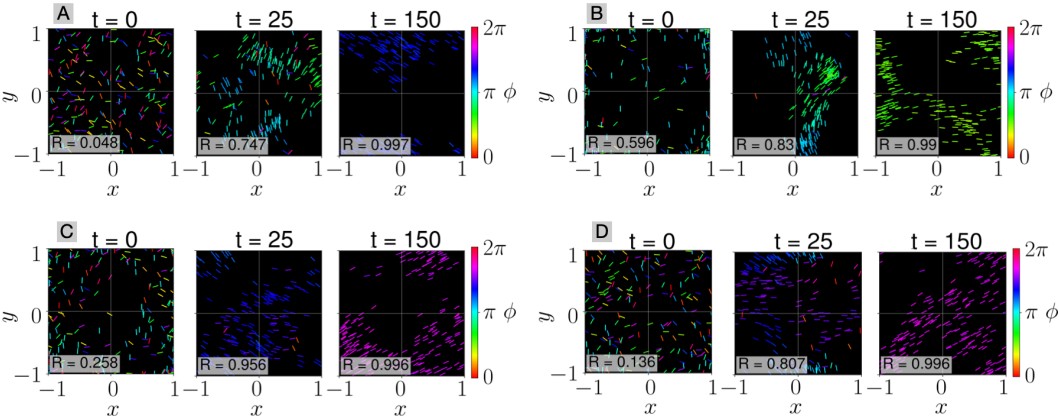

Figure 18: Open-loop behavior by using the lower-level decision policy at time $t = 0$ for all times, on Vicsek (torus) with $N = 200$ agents and various initializations. A: uniform initialization; B: beta-2 initialization; C: beta-1 initialization; D: beta-3 initialization.

Analogously, we consider some more strongly concentrated and heterogeneous initializations: The **peak-normal** initialization is given by a more concentrated zero-centered diagonal Gaussian with covariance $\sigma^2 = 0.1$. The **squeezed-normal** is the same initialization as in training, except for

dividing the variance in the $y$-axis by 10. The **multiheaded-normal** initialization is a mixture of two equisized Gaussian distributions in the upper-right and lower-left quadrant, where in comparison to the training initialization, position variances are halved. Finally, the **bernoulli-multiheaded-normal** additionally changes the weights of two Gaussians to $0.75$ and $0.25$ respectively.

As seen in Figure 19, the lower-level policy $\check{\pi}_0$ for Gaussian initialization from training easily transfers and generalizes to more complex initializations. However, the behavior may naturally be more suboptimal due to the training process likely never seeing more strongly concentrated and heterogeneous distributions of agents. For example, in the peak-normal initialization in Figure 19, we see that the agents begin relatively aligned, but will first misalign in order to align again, as the learned policy was trained to handle only the wider Gaussian initialization.

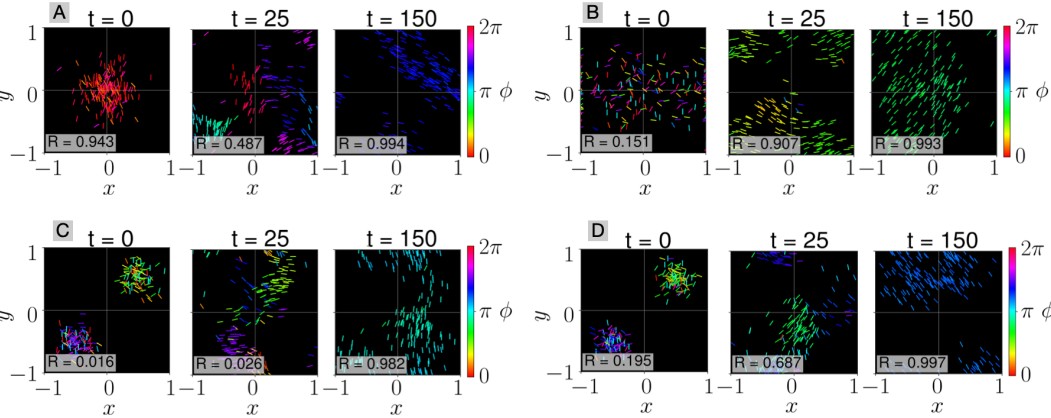

Figure 19: Open-loop behavior by using the lower-level decision policy at time $t = 0$ for all times, on Vicsek (torus) with $N = 200$ agents and various initializations. A: peak-normal initialization; B: squeezed-normal initialization; C: multiheaded-normal initialization; D: bernoulli-multiheaded-normal initialization.

**No observations.** As an additional verification of the positive effect of mean field guidance on PG training, we also perform experiments for training PPO without any RL observations, as in the previous paragraph we verified the applicability of learned behavior even without observing the MF that is observed by RL during training. In Figure 20 we see that PPO is unable to learn useful behavior, despite the existence of such a time-invariant lower-level policy from the preceding paragraph, underlining the empirical importance of mean field guidance that we derived.

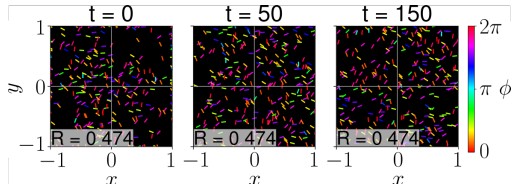

Figure 20: Qualitative behavior after training *without observations* for Vicsek (torus).

**Transfer to differing agent counts.** In Figure 21, we see qualitatively that the behavior learned for $N = 200$ agents transfers to different, lower numbers of agents as well. The result is congruent with the results shown in the main text, such as in Figure 5, and further supports the fact that our method scales to nearly arbitrary numbers of agents.

**Forward velocity control.** We also allow agents to alternatively control their maximum velocity in the range $[0, v_0]$. Forward velocity can similarly be controlled, and allows for more uniform spreading of agents in contrast to the case where velocity cannot be controlled. This shows some additional

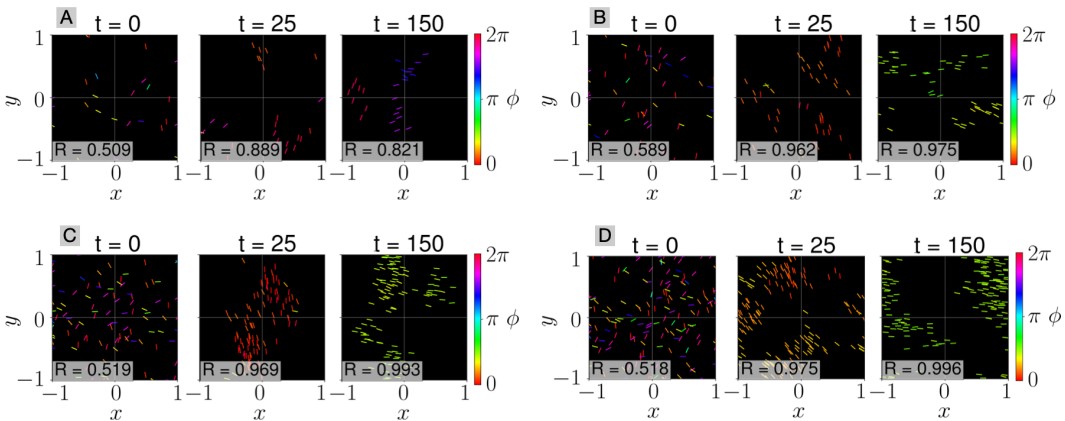

Figure 21: Qualitative behavior of the policy learned for $N = 200$ on Vicsek (torus), transferred to different numbers of agents $N$. A: $N = 25$; B: $N = 50$; C: $N = 100$; D: $N = 150$.

generalization of our algorithm to variants of collective behavior problems. The corresponding final qualitative behavior is depicted in Figure 22.

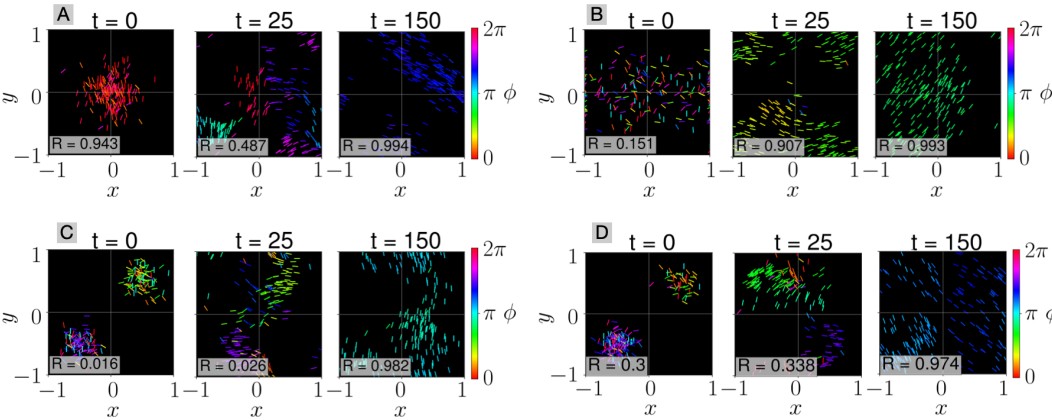

Figure 22: Qualitative behavior on Vicsek (torus) with $N = 200$ agents, additional forward velocity control, and various initializations. A: peak-normal initialization; B: squeezed-normal initialization; C: multiheaded-normal initialization; D: bernoulli-multiheaded-normal initialization.

**Comparison of IPPO and MAPPO for low numbers of agents.** Lastly, for completeness we show the comparison of IPPO and MAPPO training results for various numbers of agents in Figures 23 and 24. The overall achieved performances are overall comparable to the results of the Dec-POMFPPO method in Figure 5.

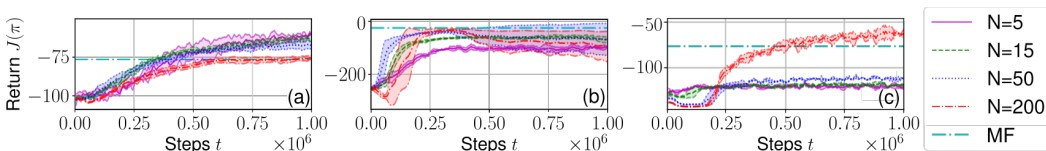

Figure 23: IPPO training curves (episode return) with shaded standard deviation over 3 seeds and various $N$, in (a) Aggregation (box), (b) Vicsek (torus), (c) Kuramoto (torus). For comparison, we also plot the best return averaged over 3 seeds for Dec-POMFPPO in Figure 3 (MF).

Figure 24: MAPPO training curves (episode return) with shaded standard deviation over 3 seeds and various $N$, in (a) Aggregation (box), (b) Vicsek (torus), (c) Kuramoto (torus). For comparison, we also plot the best return averaged over 3 seeds for Dec-POMFPPO in Figure 3 (MF).

## B  EXPERIMENTAL DETAILS

We use the RLlib 2.0.1 (Apache-2.0 license) (Liang et al., 2018) implementation of PPO (Schulman et al., 2017) for both MARL via IPPO, and our Dec-POMFPPO. For MAPPO, we used the MARLlib 1.0.0 framework (Hu et al., 2023), which builds upon RLlib. For our experiments, we used no GPUs and around $60\,000$ Intel Xeon Platinum 9242 CPU core hours, and each training run usually took at most three days by training on up to 96 CPU cores. Implementation-wise, for the upper-level policy NNs learned by PPO, we use two hidden layers with 256 nodes and $\tanh$ activations, parametrizing diagonal Gaussians over the MDP actions $\xi \in \Xi$ (parametrizations of lower-level policies).

In Aggregation, we define the parameters $\xi \in \Xi$ for continuous spaces $\mathcal{X}, \mathcal{Y}, \mathcal{U} \subseteq \mathbb{R}^d$ by values in $\Xi := [-1, 1]^{2d}$. Each component of $\xi$ is then mapped affinely to mean in $\mathcal{U}$ or diagonal covariance in $[\epsilon, 0.5 + \epsilon]$ with $\epsilon = 10^{-10}/4$, of each dimension. Meanwhile, in Vicsek and Kuramoto, we pursue a "discrete action space" approach, letting $\Xi := [-1, 1]^3$. We then affinely map components of $\xi \in \Xi$ to $[\epsilon, 0.5 + \epsilon]$, which are normalized to constitute probabilities of actions in $\{-1, 0, 1\} \subseteq \mathcal{U}$.

For the kernel-based representation of mean fields in $d$-dimensional state spaces $\mathcal{X}$, we define points $x_b$ by the center points of a $d$-dimensional gridding of spaces via equisized ($M_{\mathcal{X}} = 5^d$ hypercubes) partitions. For the histogram, we similarly use the equisized hypercube partitions. For observation spaces $\mathcal{Y}$ and the kernel-based representation of lower-level policies, unless noted otherwise (e.g. in the high-dimensional experiments below, where we use less than exponentially many points), we do the same but additionally rescale the center points $\tilde{y}_b$ around zero, giving $y_b = c\tilde{y}_b$ for some $c > 0$ and $M_{\mathcal{Y}} = 5^d$. We use $c = 0.75$ for Aggregation and $c = 0.1$ for Vicsek and Kuramoto. For the (diagonal) bandwidths of RBF kernels, in Aggregation we use $\sigma = 0.12/\sqrt{M_{\mathcal{X}}}$ for states and $\sigma = 0.12c$ for observations. In Vicsek and Kuramoto, we use $\sigma = 0.12/\sqrt{2}$ for state positions, $\sigma = 0.12\pi$ for state angles, and $\sigma = 0.06c$ or $\sigma = 0.12\pi c$ for the first or second component of observations respectively. For IPPO and MAPPO, we observe the observations $y_t$ directly. For hyperparameters of PPO, see Table 2.

Table 2: PPO hyperparameter values.

| Hyperparameter | Value |
|---|---|
| Discount factor $\gamma$ | 0.99 |
| GAE lambda | 1 |
| KL coefficient | 0.03 |
| Clip parameter | 0.2 |
| Learning rate | 0.00005 |
| Training batch size $B_{\text{len}}$ | 4000 |
| Mini-batch size $b_{\text{len}}$ | 1000 |
| Steps per batch $N_{\text{PPO}}$ | 5 |

## C  PROBLEM DETAILS

In this section, we will discuss in more detail the problems considered in our experiments.

**Aggregation.**  The Aggregation problem is a problem where agents must aggregate into a single point. Here, $\mathcal{X} = \mathcal{Y} = [-1, 1]^d \subseteq \mathbb{R}^d$ for some dimensionality parameter $d \in \mathbb{N}$, and analogously

$\mathcal{U} = [-1, 1]^d$ for per-dimension movement actions. Observations are the own, noisily observed position, and movements are similarly noisy, using Gaussian noise. Overall, the dynamics are given by

$$y_t \sim \mathcal{N}\left(x_t, \mathrm{diag}(\sigma_y^2, \dots \sigma_y^2)\right),$$

$$x_{t+1} \sim \mathcal{N}\left(x_t + v_0 \frac{u_t}{\max(1, \|u_t\|_2)}, \mathrm{diag}(\sigma_x^2, \dots \sigma_x^2)\right)$$

for some velocity $v_0$, where additionally, observations and states that are outside of the box $[-1, 1]^d$ are projected back to the box.

The reward function for aggregation of agents is defined as

$$r(\mu_t) = -c_d \iint \|x - y\|_1 \mu_t(\mathrm{d}x)\mu_t(\mathrm{d}y) - c_u \iint \left\|\frac{u}{\max(1, \|u\|_2)}\right\|_1 \pi_t(\mathrm{d}u \mid x)\mu_t(\mathrm{d}x),$$

for some disaggregation cost $c_d > 0$ and action cost $c_u > 0$, where we allow the dependence of rewards on actions as well: Note that our framework still applies to the above dependence on actions, as discussed in Section 2, by rewriting the system in the following way. At any even time $2t$, the agents transition from state $x \in \mathcal{X}$ to state-actions $(x, u) \in \mathcal{X} \times \mathcal{U}$, which will constitute the states of the new system. At the following odd times $2t+1$, the transition is sampled for the given state-actions. In this way, the mean field is over $\mathcal{X} \cup (\mathcal{X} \times \mathcal{U})$ and allows description of dependencies on the state-action distributions instead of only the state distribution.

For the experiments, we use $\sigma_x^2 = 0.04$, $\sigma_y^2 = 0.04$ and $v_0 = 0.1$. The initial distribution of agent positions is a Gaussian centered around zero, with variance $0.4$. The cost coefficients are $c_d = 1$ and $c_u = 0.1$. For simulation purposes, we consider episodes with length $T = 100$.

**Vicsek.** In classical Vicsek models, each agent is coupled to every other agent within a predefined interaction region. The agents have a fixed maximum velocity $v_0 > 0$, and attempt to align themselves with the neighboring particles within their interaction range $D > 0$. The equations governing the dynamics of the $i$-th agent in the classical Vicsek model are given in continuous time by

$$\mathrm{d}p^i = (v_0 \sin(\phi^i), v_0 \cos(\phi^i))^T \mathrm{d}t$$

$$\mathrm{d}\phi^i = \frac{1}{|N_i|} \sum_{j \in N_i} \sin\left(\phi^j - \phi^i\right) \mathrm{d}t + \sigma \mathrm{d}W$$

for all agents $i$, where $N_i$ denotes the set of agents within the interaction region, $N_i := \{j \in [N] \mid d(x^i, x^j) \leq D\}$, and $W$ denotes Brownian motion.

We consider a discrete-time variant where agents may *control* independently how to adjust their angles in order to achieve a global objective (e.g. alignment, misalignment, aggregation). For states $x_t \equiv (p_t, \phi_t)$, actions $u_t$ and observations $y_t$, we have

$$(\bar{x}, \bar{y})^T = \left(\iint \sin(\phi - \phi_t)\mathbf{1}_{d(p_t, p) \leq D}\mu_t(\mathrm{d}p, \mathrm{d}\phi), \iint \cos(\phi - \phi_t)\mathbf{1}_{d(p_t, p) \leq D}\mu_t(\mathrm{d}p, \mathrm{d}\phi)\right)^T,$$

$$y_t = \left(\|(\bar{x}, \bar{y})^T\|_2, \mathrm{atan2}\,(\bar{x}, \bar{y})\right)^T,$$

$$p_{t+1} = (p_{t,1} + v_0 \sin(\phi_t), p_{t,2} + v_0 \cos(\phi_t))^T,$$

$$\phi_{t+1} \sim \mathcal{N}(\phi_t + \omega_0 u_t, \sigma_\phi^2)$$

for some maximum angular velocity $\omega_0 > 0$ and noise covariance $\sigma_\phi^2 > 0$, where $\mathrm{atan2}(x, y)$ is the angle from the positive $x$-axis to the vector $(x, y)^T$. Therefore, we have $\mathcal{X} = [-1, 1]^2 \times [0, 2\pi)$, where positions are equipped with the corresponding topologies discussed in Appendix A, and standard Euclidean spaces $\mathcal{Y} = [-1, 1]^2$ and $\mathcal{U} = [-1, 1]$. Importantly, agents only observe the relative headings of other agents within the interaction region. As a result, it is impossible to model such a system using standard MFC techniques.

As cost functions, we consider rewards via the polarization, plus action cost as in Aggregation. Defining polarization similarly to e.g. (Zapotecatl et al., 2016),

$$\mathrm{pol}_t := \iint \angle(x, \bar{x}_t)\mu_t(\mathrm{d}p, \mathrm{d}\phi),$$

$$\angle(x, y) := \arccos\left((\cos(\phi), \sin(\phi))^T \cdot \frac{y}{\|y\|_2}\right),$$

$$\bar{x}_t := \iint (\cos(\phi), \sin(\phi))^T \mu_t(\mathrm{d}p, \mathrm{d}\phi)$$

where high values of $\mathrm{pol}_t$ indicate misalignment, we define the rewards for alignment

$$r(\mu_t) = -c_a \mathrm{pol}_t - c_u \iint |u| \pi_t(\mathrm{d}u \mid x) \mu_t(\mathrm{d}x),$$

and analogously for misalignment

$$r(\mu_t) = +c_a \mathrm{pol}_t - c_u \iint |u| \pi_t(\mathrm{d}u \mid x) \mu_t(\mathrm{d}x).$$

For our training, unless noted otherwise, we let $D = 0.25$, $v_0 = 0.075$, $\omega_0 = 0.2$, $\sigma_\phi = 0.02$ and $\mu_0$ as a zero-centered (clipped) diagonal Gaussian with variance $0.4$. The cost coefficients are $c_a = 1$ and $c_u = 0.1$. For simulation purposes, we consider episodes with length $T = 200$.

**Kuramoto.** The Kuramoto model can be obtained from the Vicsek model by setting the maximal velocity $v_0$ of the above equations to zero. Hence, we obtain a random geometric graph, where agents see only their neighbor's state distribution within the interaction region, and the neighbors are static per episode. For parameters, we let $D = 0.25$, $v_0 = 0$, $\omega_0 = 0.2$, $\sigma_\phi = 0$ and $\mu_0$ as a zero-centered (clipped) Gaussian with variance $0.4$. The cost coefficients are $c_a = 1$ and $c_u = 0.1$. For simulation purposes, we consider episodes with length $T = 200$.

## D   PROPAGATION OF CHAOS

*Proof of Theorem 1.* As in the main text, we usually equip $\mathcal{P}(\mathcal{X})$ with the 1-Wasserstein distance. In the proof, however, it is useful to also consider the uniformly equivalent metric $d_\Sigma(\mu, \mu') := \sum_{m=1}^\infty 2^{-m} |\int f_m \, \mathrm{d}(\mu - \mu')|$ instead. Here, $(f_m)_{m \geq 1}$ is a fixed sequence of continuous functions $f_m \colon \mathcal{X} \to [-1, 1]$, see e.g. (Parthasarathy, 2005, Theorem 6.6) for details.

First, let us define the measure $\zeta_t^{\pi,\mu}$ on $\mathcal{X} \times \mathcal{U}$, defined for any measurable set $A \times B \subseteq \mathcal{X} \times \mathcal{U}$ by $\zeta_t^{\pi,\mu}(A \times B) := \int_A \int_\mathcal{Y} \int_B \pi_t(\mathrm{d}u \mid y) P^y(\mathrm{d}y \mid x, \mu_t) \mu_t(\mathrm{d}x)$. For notational convenience we define the MF transition operator $\tilde{T}$ such that

$$\tilde{T}(\mu_t, \zeta_t^{\pi,\mu}) := \iint P(\cdot \mid x, u, \mu_t) \zeta_t^{\pi,\mu}(\mathrm{d}x, \mathrm{d}u) = \mu_{t+1}. \tag{6}$$

Continuity of $\tilde{T}$ follows immediately from Assumption 1a and (Cui et al., 2023, Lemma 2.5) which we recall here for convenience.

**Proposition 4** ((Cui et al., 2023), Lemma 2.5). *Under Assumption 1a, $(\mu_n, \zeta_n) \to (\mu, \zeta)$ implies $\tilde{T}(\mu_n, \zeta_n) \to \tilde{T}(\mu, \zeta)$.*

The rest of the proof is similar to (Cui et al., 2023, Theorem 2.7) – though we remark that we strengthen the convergence statement from weak convergence to convergence in $L_1$ uniformly over $f \in \mathcal{F}$ – by showing via induction over $t$ that

$$\sup_{\pi \in \Pi} \sup_{f \in \mathcal{F}} \mathbb{E}\left[|f(\mu_t^N) - f(\mu_t)|\right] \to 0. \tag{7}$$

Note that the induction start can be verified by a weak LLN argument which is also leveraged in the subsequent induction step. For the induction step we assume that (7) holds at time $t$. At time $t + 1$ we have

$$\sup_{\pi \in \Pi} \sup_{f \in \mathcal{F}} \mathbb{E}\left[|f(\mu_{t+1}^N) - f(\mu_{t+1})|\right]$$

$$\leq \sup_{\pi \in \Pi} \sup_{f \in \mathcal{F}} \mathbb{E}\left[\left|f(\mu_{t+1}^N) - f\left(\tilde{T}\left(\mu_t^N, \zeta_t^{\pi,\mu^N}\right)\right)\right|\right] \tag{8}$$

$$+ \sup_{\pi \in \Pi} \sup_{f \in \mathcal{F}} \mathbb{E}\left[\left|f\left(\tilde{T}\left(\mu_t^N, \zeta_t^{\pi,\mu^N}\right)\right) - f(\mu_{t+1})\right|\right]. \tag{9}$$

We start by analyzing the first term and recall that a modulus of continuity $\omega_{\mathcal{F}}$ of $\mathcal{F}$ is defined as a function $\omega_{\mathcal{F}}\colon [0,\infty) \to [0,\infty)$ with both $\lim_{x\to 0}\omega_{\mathcal{F}}(x) = 0$ and $|f(\mu) - f(\nu)| \le \omega_{\mathcal{F}}(W_1(\mu,\nu)), \forall f \in \mathcal{F}$. By (DeVore & Lorentz, 1993, Lemma 6.1), such a non-concave and decreasing modulus $\omega_{\mathcal{F}}$ exists for $\mathcal{F}$ because it is uniformly equicontinuous due to the compactness of $\mathcal{P}(\mathcal{X})$. Analogously, we have that $\mathcal{F}$ is uniformly equicontinuous in the space $(\mathcal{P}(\mathcal{X}), d_\Sigma)$ as well. Recalling that $\mathcal{P}(\mathcal{X})$ is compact and the topology of weak convergence is metrized by both $d_\Sigma$ and $W_1$, we know that the identity map $\mathrm{id}\colon (\mathcal{P}(\mathcal{X}), d_\Sigma) \to (\mathcal{P}(\mathcal{X}), W_1)$ is uniformly continuous. Leveraging the above findings, we have that for the identity map there exists a modulus of continuity $\tilde{\omega}$ such that

$$|f(\mu) - f(\nu)| \le \omega_{\mathcal{F}}(W_1(\mathrm{id}\,\mu, \mathrm{id}\,\nu)) \le \omega_{\mathcal{F}}(\tilde{\omega}(d_\Sigma(\mu,\nu)))$$

holds for all $\mu,\nu \in (\mathcal{P}(\mathcal{X}), d_\Sigma)$. By (DeVore & Lorentz, 1993, Lemma 6.1), we can use the least concave majorant of $\tilde{\omega}_{\mathcal{F}} := \omega_{\mathcal{F}} \circ \tilde{\omega}$ instead of $\tilde{\omega}_{\mathcal{F}}$ itself. Then, (8) can be bounded by

$$\mathbb{E}\left[\left|f(\mu_{t+1}^N) - f\left(\tilde{T}\left(\mu_t^N, \zeta_t^{\pi,\mu^N}\right)\right)\right|\right] \le \mathbb{E}\left[\tilde{\omega}_{\mathcal{F}}\left(d_\Sigma\left(\mu_{t+1}^N, \tilde{T}\left(\mu_t^N, \zeta_t^{\pi,\mu^N}\right)\right)\right)\right]$$
$$\le \tilde{\omega}_{\mathcal{F}}\left(\mathbb{E}\left[d_\Sigma\left(\mu_{t+1}^N, \tilde{T}\left(\mu_t^N, \zeta_t^{\pi,\mu^N}\right)\right)\right]\right)$$

irrespective of both $\pi$ and $f$ by the concavity of $\tilde{\omega}_{\mathcal{F}}$ and Jensen's inequality. For notational convenience, we define $x_t^N := (x_t^{i,N})_{i\in[N]}$, and arrive at

$$\mathbb{E}\left[d_\Sigma\left(\mu_{t+1}^N, \tilde{T}\left(\mu_t^N, \zeta_t^{\pi,\mu^N}\right)\right)\right] = \sum_{m=1}^{\infty} 2^{-m}\, \mathbb{E}\left[\left|\int f_m\, \mathrm{d}\left(\mu_{t+1}^N - \tilde{T}\left(\mu_t^N, \zeta_t^{\pi,\mu^N}\right)\right)\right|\right]$$
$$\le \sup_{m \ge 1} \mathbb{E}\left[\mathbb{E}\left[\left|\int f_m\, \mathrm{d}\left(\mu_{t+1}^N - \tilde{T}\left(\mu_t^N, \zeta_t^{\pi,\mu^N}\right)\right)\right| \,\middle|\, x_t^N\right]\right].$$

Finally, we require the aforementioned weak LLN argument which goes as follows

$$\mathbb{E}\left[\left|\int f_m\, \mathrm{d}\left(\mu_{t+1}^N - \tilde{T}\left(\mu_t^N, \zeta_t^{\pi,\mu^N}\right)\right)\right| \,\middle|\, x_t^N\right]^2$$
$$= \mathbb{E}\left[\left|\frac{1}{N}\sum_{i\in[N]}\left(f_m(x_{t+1}^i) - \mathbb{E}\left[f_m(x_{t+1}^i) \mid x_t^N\right]\right)\right| \,\middle|\, x_t^N\right]^2$$
$$\le \mathbb{E}\left[\left|\frac{1}{N}\sum_{i\in[N]}\left(f_m(x_{t+1}^i) - \mathbb{E}\left[f_m(x_{t+1}^i) \mid x_t^N\right]\right)\right|^2 \,\middle|\, x_t^N\right]$$
$$= \frac{1}{N^2}\sum_{i\in[N]}\mathbb{E}\left[\left(f_m(x_{t+1}^i) - \mathbb{E}\left[f_m(x_{t+1}^i) \mid x_t^N\right]\right)^2 \,\middle|\, x_t^N\right] \le \frac{4}{N} \to 0.$$

Here, we have used that $|f_m| \le 1$, as well as the conditional independence of $x_{t+1}^i$ given $x_t^N$. In combination with the above results, the term (8) thus converges to zero. Moving on to the remaining second term (9), we note that the induction assumption implies that

$$\sup_{\pi \in \Pi}\sup_{f \in \mathcal{F}} \mathbb{E}\left[\left|f\left(\tilde{T}\left(\mu_t^N, \zeta_t^{\pi,\mu^N}\right)\right) - f(\mu_{t+1})\right|\right]$$
$$= \sup_{\pi \in \Pi}\sup_{f \in \mathcal{F}} \mathbb{E}\left[\left|f\left(\tilde{T}\left(\mu_t^N, \zeta_t^{\pi,\mu^N}\right)\right) - f\left(\tilde{T}\left(\mu_t, \zeta_t^{\pi,\mu}\right)\right)\right|\right]$$
$$\le \sup_{\pi \in \Pi}\sup_{g \in \mathcal{G}} \mathbb{E}\left[\left|g(\mu_t^N) - g(\mu_t)\right|\right] \to 0$$

using the function $g := f \circ \tilde{T}_*^{\pi_t}$ which belongs to the class $\mathcal{G}$ of equicontinuous functions with modulus of continuity $\omega_{\mathcal{G}} := \omega_{\mathcal{F}} \circ \omega_{\tilde{T}}$. Here, $\omega_{\tilde{T}}$ is the uniform modulus of continuity over all policies $\pi$ of $\mu_t \mapsto \tilde{T}_*^{\pi_t}(\mu_t) := \tilde{T}(\mu_t, \zeta_t^{\pi,\mu})$. The equicontinuity of $\{\tilde{T}_*^{\pi_t}\}_{\pi \in \Pi}$ is a consequence of Lemma 4 as well as the equicontinuity of functions $\mu_t \mapsto \zeta_t^{\pi,\mu}$ which in turn follows from the

uniform Lipschitzness of $\Pi$. The validation of this claim is provided in the next lines. Note that this also completes the induction and thereby the proof. For a sequence of $\mu_n \to \mu \in \mathcal{P}(\mathcal{X})$ we can write

$$\sup_{\pi \in \Pi} W_1(\zeta_t^{\pi,\mu_n}, \zeta_t^{\pi,\mu})$$

$$\leq \sup_{\pi \in \Pi} \sup_{\|f'\|_{\mathrm{Lip}} \leq 1} \left| \iiint f'(x,u) \pi_t(\mathrm{d}u \mid y)(P^y(\mathrm{d}y \mid x, \mu_n) - P^y(\mathrm{d}y \mid x, \mu))\mu_n(\mathrm{d}x) \right|$$

$$+ \sup_{\pi \in \Pi} \sup_{\|f'\|_{\mathrm{Lip}} \leq 1} \left| \iiint f'(x,u) \pi_t(\mathrm{d}u \mid y) P^y(\mathrm{d}y \mid x, \mu)(\mu_n(\mathrm{d}x) - \mu(\mathrm{d}x)) \right|.$$

Starting with the first term, we apply Assumptions 1a–1b to arrive at

$$\sup_{\pi \in \Pi} \sup_{\|f'\|_{\mathrm{Lip}} \leq 1} \left| \iiint f'(x,u) \pi_t(\mathrm{d}u \mid y)(P^y(\mathrm{d}y \mid x, \mu_n) - P^y(\mathrm{d}y \mid x, \mu))\mu_n(\mathrm{d}x) \right|$$

$$\leq \sup_{\pi \in \Pi} \sup_{\|f'\|_{\mathrm{Lip}} \leq 1} \int \left| \iint f'(x,u) \pi_t(\mathrm{d}u \mid y)(P^y(\mathrm{d}y \mid x, \mu_n) - P^y(\mathrm{d}y \mid x, \mu)) \right| \mu_n(\mathrm{d}x)$$

$$\leq \sup_{\pi \in \Pi} \sup_{\|f'\|_{\mathrm{Lip}} \leq 1} \sup_{x \in \mathcal{X}} \left| \iint f'(x,u) \pi_t(\mathrm{d}u \mid y)(P^y(\mathrm{d}y \mid x, \mu_n) - P^y(\mathrm{d}y \mid x, \mu)) \right|$$

$$\leq L_\Pi \sup_{x \in \mathcal{X}} W_1(P^y(\cdot \mid x, \mu_n), P^y(\cdot \mid x, \mu))$$

$$\leq L_\Pi L_{P^y} W_1(\mu_n, \mu) \to 0$$

with Lipschitz constant $L_\Pi$ corresponding to the Lipschitz function $y \mapsto \int f'(x,u) \pi_t(\mathrm{d}u \mid y)$. Alternatively, if $P^y$ is assumed independent of the mean field in Assumption 1b, the term is zero.

In a similar fashion, we point out the 1-Lipschitzness of $x \mapsto \iint \frac{f'(x,u)}{L_\Pi L_{P^y}+1} \pi_t(\mathrm{d}u \mid y) P^y(\mathrm{d}y \mid x, \mu)$, as

$$\left| \iint \frac{f'(z,u)}{L_\Pi L_{P^y}+1} \pi_t(\mathrm{d}u \mid y) P^y(\mathrm{d}y \mid z, \mu) - \iint \frac{f'(x,u)}{L_\Pi L_{P^y}+1} \pi_t(\mathrm{d}u \mid y) P^y(\mathrm{d}y \mid x, \mu) \right|$$

$$\leq \left| \iint \frac{f'(z,u) - f'(x,u)}{L_\Pi L_{P^y}+1} \pi_t(\mathrm{d}u \mid y) P^y(\mathrm{d}y \mid z, \mu) \right|$$

$$+ \left| \iint \frac{f'(x,u)}{L_\Pi L_{P^y}+1} \pi_t(\mathrm{d}u \mid y)(P^y(\mathrm{d}y \mid z, \mu) - P^y(\mathrm{d}y \mid x, \mu)) \right|$$

$$\leq \frac{1}{L_\Pi L_{P^y}+1} d(z,x) + \frac{L_\Pi}{L_\Pi L_{P^y}+1} W_1(P^y(\mathrm{d}y \mid z, \mu), P^y(\mathrm{d}y \mid x, \mu))$$

$$\leq \left( \frac{1}{L_\Pi L_{P^y}+1} + \frac{L_\Pi L_{P^y}}{L_\Pi L_{P^y}+1} \right) d(x,y) = d(x,y)$$

for $z \neq x$. Alternatively, if the state space is assumed finite in Assumption 1b, the Lipschitzness follows directly.

This eventually yields the convergence of the second term, i.e.

$$\sup_{\pi \in \Pi} \sup_{\|f'\|_{\mathrm{Lip}} \leq 1} \left| \iiint f'(x,u) \pi_t(\mathrm{d}u \mid y) P^y(\mathrm{d}y \mid x, \mu)(\mu_n(\mathrm{d}x) - \mu(\mathrm{d}x)) \right|$$

$$= \sup_{\pi \in \Pi} \sup_{\|f'\|_{\mathrm{Lip}} \leq 1} (L_{P^y} L_\Pi + 1) \left| \iint \frac{f'(x,u)}{L_{P^y} L_\Pi + 1} \pi_t(\mathrm{d}u \mid y) P^y(\mathrm{d}y \mid x, \mu)(\mu_n(\mathrm{d}x) - \mu(\mathrm{d}x)) \right|$$

$$\leq (L_{P^y} L_\Pi + 1) W_1(\mu_n, \mu) \to 0$$

and thus completes the proof. $\qquad \square$

In the special case of finite states and actions, the approximation rate can also be quantified to $\mathcal{O}(1/\sqrt{N})$ by considering equi-Lipschitz families of functions $\mathcal{F}$ with constant $L_f$. Then, there is no need to consider the two different metrizations $d_\Sigma$ and $W_1$, as they are Lipschitz equivalent, and one

can simply use the $L_1$ distance. The convergence in the first term (8) is then directly via the weak LLN at rate $\mathcal{O}(1/\sqrt{N})$ by

$$\sup_{\pi \in \Pi} \sup_{f \in \mathcal{F}} \mathbb{E}\left[\left|f(\mu_{t+1}^N) - f\left(\tilde{T}\left(\mu_t^N, \zeta_t^{\pi,\mu^N}\right)\right)\right|\right]$$

$$\leq \sup_{\pi \in \Pi} L_f \, \mathbb{E}\left[\sum_{x \in \mathcal{X}} \left|\mu_{t+1}^N(x) - \tilde{T}\left(\mu_t^N, \zeta_t^{\pi,\mu^N}\right)(x)\right|\right]$$

$$= \sup_{\pi \in \Pi} L_f \sum_{x \in \mathcal{X}} \mathbb{E}\left[\mathbb{E}\left[\left|\frac{1}{N}\sum_{i=1}^N \mathbf{1}_x(x_{t+1}^{i,N}) - \mathbb{E}\left[\frac{1}{N}\sum_{i=1}^N \mathbf{1}_x(x_{t+1}^{i,N}) \,\Big|\, x_t^N\right]\right| \,\Big|\, x_t^N\right]\right]$$

$$\leq L_f |\mathcal{X}| \sqrt{\frac{4}{N}},$$

while for the second term (9) we use the induction assumption, since $\tilde{T}$ is uniformly Lipschitz.

## E  AGENTS WITH MEMORY AND HISTORY-DEPENDENCE

For agents with bounded memory, we note that such memory can be analyzed by our model by adding the memory state to the usual agent state, and manipulations on the memory either to the actions or transition dynamics.

For example, let $z_t^i \in \mathcal{Q} := \{0,1\}^Q$ be the $Q$-bit memory of an agent at any time. Then, we may consider the new $\mathcal{X} \times \mathcal{Q}$-valued state $(x_t^i, z_t^i)$, which remains compact, and the new $\mathcal{U} \times \mathcal{Q}$-valued actions $(u_t^i, w_t^i)$, where $w_t^i$ is a write action that can arbitrarily rewrite the memory, $z_{t+1}^i = w_t^i$. Theoretical properties are preserved by discreteness of added states and actions.

Analogously, extending transition dynamics to include observations $y$ also allows for description of history-dependent policies. This approach extends to infinite-memory states, by adding observations $y$ also to the transition dynamics, and considering histories for states and observations. Define the observation space of histories $\mathcal{Y}' := \mathcal{Y} \times \bigcup_{i=0}^\infty (\mathcal{Y} \times \mathcal{U})^i$, and the according state space $\mathcal{X}' := \mathcal{X} \times \bigcup_{i=0}^\infty (\mathcal{Y} \times \mathcal{U})^i$. The new mean fields $\mu_t^N, \mu_t$ are thus $\mathcal{P}(\mathcal{X}')$-valued. The new observation-dependent dynamics are then defined by

$$P'(\cdot \mid x, y, u, \mu) = P(\cdot \mid x_1, u, \mathrm{marg}_1 \mu) \otimes \delta_{(\mathbf{x}_2, y, u)}$$

where $\mathrm{marg}_1$ maps $\mu$ to its first marginal, $x_1$ is the first component of $x$, and $\mathbf{x}_2$ is the $(\mathcal{Y} \times \mathcal{U})^t$-valued past history. Here, $(\mathbf{x}_2, y, u)$ defines the new history of an agent, which is observed by

$$P^{y'}(\cdot \mid x, \mu) = P^y(\cdot \mid x_1, \mathrm{marg}_1 \mu) \otimes \delta_{\mathbf{x}_2}.$$

Clearly, Lipschitz continuity is preserved. Further, we obtain the mean field transition operator

$$T'(\mu_t, h_t') := \iiint P'(\cdot \mid x, y, u, \mu_t) h_t'(\mathrm{d}x, \mathrm{d}y, \mathrm{d}u).$$

using $\mathcal{X}' \times \mathcal{Y}' \times \mathcal{U}$-valued actions $h_t' = \mu_t \otimes P^y(\mu_t) \otimes \check{\pi}[h_t']$ for some Lipschitz $\check{\pi}[h_t'] \colon \mathcal{Y}' \to \mathcal{P}(\mathcal{U})$. And in particular, the proof of e.g. Theorem 1 extends to this new case. For example, the weak LLN argument still holds by

$$\mathbb{E}\left[d_\Sigma\left(\mu_{t+1}^N, T'\left(\mu_t^N, h_t'\right)\right)\right]$$

$$\leq \sup_{m \geq 1} \mathbb{E}\left[\mathbb{E}\left[\left|\int f_m \, \mathrm{d}\left(\mu_{t+1}^N - T'\left(\mu_t^N, h_t'\right)\right)\right| \,\Big|\, x_t^N\right]\right]$$

$$\leq \sup_{m \geq 1} \mathbb{E}\left[\left|\frac{1}{N}\sum_{i \in [N]} \left(f_m(x_{t+1}^i, y_0^i, u_0^i, \ldots, y_t^i, u_t^i)\right.\right.\right.$$

$$\left.\left.\left. - \mathbb{E}\left[f_m(x_{t+1}^i, y_0^i, u_0^i, \ldots, y_t^i, u_t^i) \mid x_t^N\right]\right)\right|^2 \,\Big|\, x_t^N\right]^{\frac{1}{2}} \leq \frac{4}{N} \to 0.$$

for appropriate sequences of functions $(f_m)_{m \geq 1}$, $f_m \colon \mathcal{X} \times (\mathcal{Y} \times \mathcal{U})^{t+1} \to [-1, 1]$ (Parthasarathy, 2005) and

$$\int f_m \, \mathrm{d}T' \left( \mu_t^N, h_t' \right) = \int f_m(x_{t+1}, y_0, u_0, \ldots, y_t, u_t) P'(\mathrm{d}x_{t+1} \mid x_t, y_t, u_t, \mu_t^N)$$
$$\check{\pi}[h_t'](\mathrm{d}u_t \mid y_t) P^{y'}(\mathrm{d}y_t \mid x_t, \mu_t^N) \mu_t^N(\mathrm{d}x_t, \mathrm{d}y_0, \mathrm{d}u_0, \ldots, \mathrm{d}y_{t-1}, \mathrm{d}u_{t-1}).$$

Analogously, we can see that the above is part of a set of equicontinuous functions, and again allows application of the induction assumption, completing the extension.

## F  APPROXIMATE MFC-TYPE DEC-POMDP OPTIMALITY

*Proof of Corollary 1.* The finite-agent discounted objective converges uniformly over policies to the MFC objective

$$\sup_{\pi \in \Pi} \left| J^N(\pi) - J(\pi) \right| \to 0 \quad \text{as } N \to \infty, \tag{10}$$

since for any $\varepsilon > 0$, let $T \in \mathcal{T}$ such that $\sum_{t=T}^{\infty} \gamma^t \, \mathbb{E} \left| \left[ r(\mu_t^N) - r(\mu_t) \right] \right| \leq \frac{\gamma^T}{1-\gamma} \max_\mu 2|r(\mu)| < \frac{\varepsilon}{2}$, and further let $\sum_{t=0}^{T-1} \gamma^t \, \mathbb{E} \left| \left[ r(\mu_t^N) - r(\mu_t) \right] \right| < \frac{\varepsilon}{2}$ by Theorem 1 for sufficiently large $N$.

Therefore, approximate optimality is obtained by

$$J^N(\pi) - \sup_{\pi' \in \Pi} J^N(\pi') = \inf_{\pi' \in \Pi} (J^N(\pi) - J^N(\pi'))$$
$$\geq \inf_{\pi' \in \Pi} (J^N(\pi) - J(\pi)) + \inf_{\pi' \in \Pi} (J(\pi) - J(\pi')) + \inf_{\pi' \in \Pi} (J(\pi') - J^N(\pi'))$$
$$\geq -\frac{\varepsilon}{2} + 0 - \frac{\varepsilon}{2} = -\varepsilon$$

by the optimality of $\pi \in \arg\max_{\pi' \in \Pi} J(\pi')$ and (10) for sufficiently large $N$. □

## G  EQUIVALENCE OF DEC-POMFC AND DEC-MFC

*Proof of Proposition 1.* We begin by showing the first statement. The proof is by showing $\bar{\mu}_t = \mu_t$ at all times $t \in \mathcal{T}$, as it then follows that $\bar{J}(\bar{\pi}) = \sum_{t=0}^{\infty} \gamma^t r(\bar{\mu}_t) = \sum_{t=0}^{\infty} \gamma^t r(\mu_t) = J(\pi)$. At time $t = 0$, we have by definition $\bar{\mu}_0 = \mu_0$. Assume $\bar{\mu}_t = \mu_t$ at time $t$, then at time $t + 1$, by (2) and (3), we have

$$\bar{\mu}_{t+1} = \iiint P(x, u, \mu_t) \bar{\pi}_t(\mathrm{d}u \mid y, \bar{\mu}_t) P^y(\mathrm{d}y \mid x, \bar{\mu}_t) \bar{\mu}_t(\mathrm{d}x) \tag{11}$$
$$= \iiint P(x, u, \mu_t) \pi_t(\mathrm{d}u \mid y) P^y(\mathrm{d}y \mid x, \mu_t) \mu_t(\mathrm{d}x) = \mu_{t+1} \tag{12}$$

which is the desired statement. An analogous proof for the second statement completes the proof. □

## H  OPTIMALITY OF DEC-MFC SOLUTIONS

*Proof of Corollary 2.* Assume $J(\Phi(\bar{\pi})) < \sup_{\pi' \in \Pi} J(\pi')$. Then there exists $\pi' \in \Pi$ such that $J(\Phi(\bar{\pi})) < J(\pi')$. But by Proposition 1, there exists $\bar{\pi}' \in \bar{\Pi}$ such that $\bar{J}(\bar{\pi}') = J(\pi')$ and hence $\bar{J}(\bar{\pi}) = J(\Phi(\bar{\pi})) < J(\pi') = \bar{J}(\bar{\pi}')$, which contradicts $\bar{\pi} \in \arg\max_{\bar{\pi}' \in \bar{\Pi}} \bar{J}(\bar{\pi}')$. Therefore, $\Phi(\bar{\pi}) \in \arg\max_{\pi' \in \Pi} J(\pi')$. □

## I  DYNAMIC PROGRAMMING PRINCIPLE

*Proof of Theorem 2.* We verify the assumptions in (Hernández-Lerma & Muñoz de Ozak, 1992). First, note the (weak) continuity of transition dynamics $\hat{T}$.

**Proposition 5.** *Under Assumption 1a, $\hat{T}(\mu_n, h_n) \to \hat{T}(\mu, h)$ for any sequence $(\mu_n, h_n) \to (\mu, h)$ of MFs $\mu_n, \mu \in \mathcal{P}(\mathcal{X})$ and joint distributions $h_n \in \mathcal{H}(\mu_n)$, $h \in \mathcal{H}(\mu)$.*

*Proof.* The convergence $h_n \to h$ also implies the convergence of its marginal $\int_{\mathcal{Y}} h_n(\cdot, \mathrm{d}y, \cdot) \to \int_{\mathcal{Y}} h(\cdot, \mathrm{d}y, \cdot)$. The proposition then follows immediately from Proposition 4. ∎

Furthermore, the reward is continuous and hence bounded by Assumption 1a. It is inf-compact by

$$\{h \in \mathcal{H}(\mu) \mid -r(\mu) \le c\} = \begin{cases} \mathcal{H}(\mu) & \text{if } -r(\mu) \le c, \\ \emptyset & \text{else,} \end{cases}$$

where $\mathcal{H}(\mu)$ is closed by (Cui et al., 2023, Appendix A.2), and Lemma 2 if considering equi-Lipschitz policies in Assumption 1b.

Further, by compactness of $\mathcal{P}(\mathcal{X} \times \mathcal{Y} \times \mathcal{U})$, $\mathcal{H}(\mu)$ is compact as a closed subset of a compact set.

Lastly, lower semicontinuity of $\mu \mapsto \mathcal{H}(\mu)$ is given, since for any $\mu_n \to \mu$ and $h = \mu \otimes P^y(\mu) \otimes \check{\pi} \in \mathcal{H}(\mu)$, we can find $h_n \in \mathcal{H}(\mu_n)$: Let $h_n = \mu_n \otimes P^y(\mu_n) \otimes \check{\pi}$, then

$$W_1(h_n, h) = \sup_{f \in \mathrm{Lip}(1)} \iiint f(x, y, u) \check{\pi}(\mathrm{d}u \mid y) \left( P^y(\mathrm{d}y \mid x, \mu_n) \mu_n(\mathrm{d}x) - P^y(\mathrm{d}y \mid x, \mu) \mu(\mathrm{d}x) \right)$$

$$\le \sup_{f \in \mathrm{Lip}(1)} \iiint f(x, y, u) \check{\pi}(\mathrm{d}u \mid y) \left( P^y(\mathrm{d}y \mid x, \mu_n) - P^y(\mathrm{d}y \mid x, \mu) \right) \mu_n(\mathrm{d}x)$$

$$+ \sup_{f \in \mathrm{Lip}(1)} \iiint f(x, y, u) \check{\pi}(\mathrm{d}u \mid y) P^y(\mathrm{d}y \mid x, \mu) \left( \mu_n(\mathrm{d}x) - \mu(\mathrm{d}x) \right)$$

$$\le \sup_{f \in \mathrm{Lip}(1)} \int \left| \iint f(x, y, u) \check{\pi}(\mathrm{d}u \mid y) \left( P^y(\mathrm{d}y \mid x, \mu_n) - P^y(\mathrm{d}y \mid x, \mu) \right) \right| \mu_n(\mathrm{d}x)$$

$$+ \sup_{f \in \mathrm{Lip}(1)} \iiint f(x, y, u) \check{\pi}(\mathrm{d}u \mid y) P^y(\mathrm{d}y \mid x, \mu) \left( \mu_n(\mathrm{d}x) - \mu(\mathrm{d}x) \right) \to 0$$

since the integrands are Lipschitz by Assumption 1a and analyzed as in the proof of Theorem 1.

The proof concludes by (Hernández-Lerma & Muñoz de Ozak, 1992, Theorem 4.2). ∎

## J CONVERGENCE LEMMA

**Lemma 1.** *Assume that $(X, d)$ is a complete metric space and that $(x_n)_{n \in \mathbb{N}}$ is a sequence of elements of $X$. Then, the convergence condition of the sequence $(x_n)_{n \in \mathbb{N}}$, i.e. that*

$$\exists x \in X : \forall \varepsilon > 0 : \exists N \in \mathbb{N} : \forall n \ge N : d(x, x_n) < \varepsilon \tag{13}$$

*holds, is equivalent to the statement*

$$\forall \varepsilon > 0 : \exists x \in X : \exists N \in \mathbb{N} : \forall n \ge N : d(x, x_n) < \varepsilon. \tag{14}$$

*Proof.* (13) $\Rightarrow$ (14): follows immediately.

(14) $\Rightarrow$ (13): Choose some strictly monotonically decreasing, positive sequence of $(\varepsilon_i)_{i \in \mathbb{N}}$ with $\lim_{i \to \infty} \varepsilon_i = 0$. Then, by statement (14) we can define corresponding sequences $(x_i)_{i \in \mathbb{N}}$ and $(N_i)_{i \in \mathbb{N}}$ such that

$$\forall n \ge N_i : d(x_i, x_n) < \varepsilon_i. \tag{15}$$

Consider $i, i' \in \mathbb{N}$ and assume w.l.o.g. $i < i'$. We know by the triangle inequality

$$\forall n \ge \max\{N_i, N_{i'}\} : d(x_i, x_{i'}) \le d(x_i, x_n) + d(x_n, x_{i'}) \le 2\varepsilon_i. \tag{16}$$

Thus, the sequence $(x_i)_{i \in \mathbb{N}}$ is Cauchy and therefore converges to some $x \in X$ because $(X, d)$ is a complete metric space by assumption. Specifically, this is equivalent to

$$\exists x \in X : \forall \varepsilon > 0 : \exists I \in \mathbb{N} : \forall i \ge I : d(x, x_i) < \varepsilon. \tag{17}$$

Finally, statements (16), (17), and the triangle inequality yield

$$\exists x \in X : \forall 2\varepsilon > 0 : \exists N \in \mathbb{N} : \forall n \ge N : d(x, x_n) \le d(x, x_i) + d(x_i, x_n) < 2\varepsilon$$

which implies the desired statement (13) and concludes the proof. ∎

## K  CLOSEDNESS OF JOINT MEASURES UNDER EQUI-LIPSCHITZ KERNELS

**Lemma 2.** *Let $\mu_{xy} \in \mathcal{P}(\mathcal{X} \times \mathcal{Y})$ be arbitrary. For any $h_n = \mu_{xy} \otimes \check{\pi}_n \to h \in \mathcal{P}(\mathcal{X} \times \mathcal{Y} \times \mathcal{U})$ with $L_\Pi$-Lipschitz $\check{\pi}_n \in \mathcal{P}(\mathcal{U})^{\mathcal{Y}}$, there exists $L_\Pi$-Lipschitz $\check{\pi} \in \mathcal{P}(\mathcal{U})^{\mathcal{Y}}$ such that $h = \mu_{xy} \otimes \check{\pi}$.*

*Proof.* For readability, we write $\mu_y \in \mathcal{P}(\mathcal{Y})$ for the second marginal of $\mu_{xy}$. The required $\check{\pi}$ is constructed as the $\mu_y$-a.e. pointwise limit of $y \mapsto \check{\pi}_n(y) \in \mathcal{P}(\mathcal{U})$, as $\mathcal{P}(\mathcal{U})$ is sequentially compact under the topology of weak convergence by Prokhorov's theorem (Billingsley, 2013). For the proof, we assume Hilbert $\mathcal{Y}$ and finite actions $\mathcal{U}$, making $\mathcal{P}(\mathcal{U})$ Euclidean.

First, **(i)** we show that $\check{\pi}_n(y)$ must converge for $\mu_y$-a.e. $y \in \mathcal{Y}$ to some arbitrary limit, which we define as $\check{\pi}(y)$. It then follows by Egorov's theorem (e.g. (Kallenberg, 2021, Lemma 1.38)) that for any $\epsilon > 0$, there exists a measurable set $A \in \mathcal{Y}$ such that $\mu_y(A) < \epsilon$ and $\check{\pi}_n(y)$ converges uniformly on $\mathcal{Y} \setminus A$. Therefore, we obtain that $\check{\pi}$ restricted to $\mathcal{Y} \setminus A$ is $L_\Pi$-Lipschitz as a uniform limit of $L_\Pi$-Lipschitz functions, hence $\mu_y$-a.e. $L_\Pi$-Lipschitz. **(ii)** We then extend $\check{\pi}$ on the entire space $\mathcal{Y}$ to be $L_\Pi$-Lipschitz. **(iii)** All that remains is to show that indeed, the extended $\check{\pi}$ fulfills $h = \mu_y \otimes \check{\pi}$, which is the desired closedness.

**(i) Almost-everywhere convergence.**  To prove the $\mu_y$-a.e. convergence, we perform a proof by contradiction and assume the statement is not true. Then there exists a measurable set $A \subseteq \mathcal{Y}$ with positive measure $\mu_y(A) > 0$ such that for all $y \in A$ the sequence $\check{\pi}_n(y) \in \mathcal{P}(\mathcal{U})$ does not converge as $n \to \infty$. We show that then, $\mu_y \otimes \check{\pi}_n$ does not converge to any limiting $\tilde{h} \in \mathcal{P}(\mathcal{Y} \times \mathcal{U})$, which is a contradiction with the premise and completes the proof.

**Lemma 3.** *There exists $y^* \in A$ such that for any $r > 0$, the set $B_r(y^*) \cap A$ has positive measure.*

*Proof of Lemma 3.* Consider an open cover $\bigcup_{y \in A} B_r(y)$ of $A$ using balls $B_r$ with radius $r$, and choose a finite subcover $\{B_r(y_i)\}_{i=1,\ldots,K}$ of $A$ by compactness of $\mathcal{Y}$. Then, there exists a ball $B_r(y^*)$ from the finite subcover around a point $y^* \in \mathcal{Y}$ such that $\mu_y(B_r(y^*) \cap A) > 0$, as otherwise $\mu_y(A) = \mu_y(\bigcup_{i=1}^K B_r(y_i) \cap A) \leq \sum_{i=1}^K \mu_y(B_r(y_i) \cap A) = 0$ contradicts $\mu_y(A) > 0$.

By repeating the argument, there must exist $y^* \in A$ for which we have for any $r > 0$ that the ball $B_r(y^*) \cap A$ has positive measure. More precisely, consider a sequence of radii $r_k = 1/k$, $k \geq 1$, and repeatedly choose balls $B_{r_{k+1}} \subseteq B_{r_k}$ from an open cover of $B_{r_k} \cap A$ such that $\mu(\bar{B}_{r_{k+1}} \cap B_{r_k} \cap A) > 0$, starting with $B_{r_1} \subseteq \mathcal{Y}$ such that $\mu(B_{r_1} \cap A) > 0$. By induction, we thus have for any $k$ that $\mu(B_{r_k} \cap A) > 0$. The sequence $(B_{r_k})_{k \in \mathbb{N}}$ produces a decreasing sequence of compact sets by taking the closure of the balls $\bar{B}_{r_k}$. By Cantor's intersection theorem (Rudin, 1976, Theorem 2.36), the intersection is non-empty, $\bigcap_{k \in \mathbb{N}} \bar{B}_{r_k} \neq \emptyset$. Choose arbitrary $y^* \in \bigcap_{k \in \mathbb{N}} \bar{B}_{r_k}$, then for any $r > 0$ we have that $B_{r_k} \subseteq B_r(y^*)$ for some $k$ by $r_k \to 0$. Therefore, $\mu(B_r(y^*) \cap A) \geq \mu(B_{r_k} \cap A) > 0$. ∎

**Bounding difference to assumed limit from below.**  Choose $y^*$ according to Lemma 3. By (14) in Lemma 1, since $\check{\pi}_n(y^*) \in \mathcal{P}(\mathcal{U})$ does not converge, there exists $\epsilon > 0$ such that for all $r > 0$, infinitely often (i.o.) in $n$,

$$W_1\left(\check{\pi}_n(\cdot \mid y^*), \frac{1}{\mu_y(B_r(y^*))} \int_{B_r(y^*)} \tilde{\pi}(\cdot \mid y)\mu_y(\mathrm{d}y)\right)$$

$$= \frac{1}{2} \sum_{u \in \mathcal{U}} \left| \check{\pi}_n(u \mid y^*) - \frac{1}{\mu_y(B_r(y^*))} \int_{B_r(y^*)} \tilde{\pi}(u \mid y)\mu_y(\mathrm{d}y) \right| > \epsilon$$

where for finite $\mathcal{U}$, $W_1$ is equivalent to the total variation norm (Gibbs & Su, 2002, Theorem 4), which is half the $L_1$ norm, and $\tilde{\pi}$ is not necessarily Lipschitz and results from disintegration of $h$ into $h = \mu_y \otimes \tilde{\pi}$ (Kallenberg, 2021).

Now fix arbitrary $\epsilon' \in (\frac{\epsilon}{2}, \epsilon)$. Then, by the prequel, we define the non-empty set $\bar{\mathcal{U}}(r) \subseteq \mathcal{U}$ by excluding all actions where the absolute value is less than $\frac{\epsilon - \epsilon'}{|\mathcal{U}|}$, i.e.

$$\bar{\mathcal{U}}(r) := \left\{ u \in \mathcal{U} \;\middle|\; \left| \check{\pi}_n(u \mid y^*) - \frac{1}{\mu_y(B_r(y^*))} \int_{B_r(y^*)} \tilde{\pi}(u \mid y) \mu_y(\mathrm{d}y) \right| \geq \frac{\epsilon - \epsilon'}{|\mathcal{U}|} \right\},$$

such that

$$\frac{1}{2} \sum_{u \in \bar{\mathcal{U}}(r)} \left| \check{\pi}_n(u \mid y^*) - \frac{1}{\mu_y(B_r(y^*))} \int_{B_r(y^*)} \tilde{\pi}(u \mid y) \mu_y(\mathrm{d}y) \right| > \epsilon' \tag{18}$$

since we have the bound on the value contributed by excluded actions $u \notin \bar{\mathcal{U}}(r)$

$$\frac{1}{2} \sum_{u \notin \bar{\mathcal{U}}(r)} \left| \check{\pi}_n(u \mid y^*) - \frac{1}{\mu_y(B_r(y^*))} \int_{B_r(y^*)} \tilde{\pi}(u \mid y) \mu_y(\mathrm{d}y) \right| \leq \frac{\epsilon - \epsilon'}{2} < \epsilon - \epsilon'. \tag{19}$$

By $L_\Pi$-Lipschitz $\check{\pi}_n$, we also have for all $y \in B_r(y^*)$ that $W_1(\check{\pi}_n(y), \check{\pi}_n(y^*)) < L_\Pi r$. Hence, in particular if we choose $r = \frac{1}{L_\Pi} \min\left( \epsilon' - \frac{\epsilon}{2}, \frac{\epsilon'}{2}, \frac{\epsilon - \epsilon'}{4|\mathcal{U}|} \right)$, then for all $y \in B_r(y^*)$

$$\frac{1}{2} \sum_{u \in \mathcal{U}} |\check{\pi}_n(u \mid y) - \check{\pi}_n(u \mid y^*)| < \min\left( \epsilon' - \frac{\epsilon}{2}, \frac{\epsilon'}{2}, \frac{\epsilon - \epsilon'}{4|\mathcal{U}|} \right) \tag{20}$$

and in particular also

$$|\check{\pi}_n(u \mid y) - \check{\pi}_n(u \mid y^*)| < \frac{\epsilon - \epsilon'}{2|\mathcal{U}|}$$

for all actions $u \in \bar{\mathcal{U}}(r)$, such that by definition of $\bar{\mathcal{U}}(r)$, we find that the sign of the value inside the absolute value must not change on the entirety of $y \in B_r(y^*)$, i.e.

$$\mathrm{sgn}\left( \check{\pi}_n(u \mid y) - \frac{1}{\mu_y(B_r(y^*))} \int_{B_r(y^*)} \tilde{\pi}(u \mid y') \mu_y(\mathrm{d}y') \right)$$

$$= \mathrm{sgn}\left( \check{\pi}_n(u \mid y^*) - \frac{1}{\mu_y(B_r(y^*))} \int_{B_r(y^*)} \tilde{\pi}(u \mid y') \mu_y(\mathrm{d}y') \right)$$

which implies, since the signs must match for all $y$ with the term for $y^*$, by integrating over $y$

$$\mathrm{sgn}\left( \int_{B_r(y^*)} \check{\pi}_n(u \mid y') \mu_y(\mathrm{d}y') - \int_{B_r(y^*)} \tilde{\pi}(u \mid y') \mu_y(\mathrm{d}y') \right)$$

$$= \mathrm{sgn}\left( \int_{B_r(y^*)} \left( \check{\pi}_n(u \mid y^*) - \tilde{\pi}(u \mid y') \right) \mu_y(\mathrm{d}y') \right). \tag{21}$$

From the triangle inequality,

$$\frac{1}{2} \sum_{u \in \bar{\mathcal{U}}(r)} \left| \int_{B_r(y^*)} \left( \check{\pi}_n(u \mid y^*) - \tilde{\pi}(u \mid y) \right) \mu_y(\mathrm{d}y) \right|$$

$$\leq \frac{1}{2} \sum_{u \in \bar{\mathcal{U}}(r)} \left| \int_{B_r(y^*)} \left( \check{\pi}_n(u \mid y^*) - \check{\pi}_n(u \mid y) \right) \mu_y(\mathrm{d}y) \right|$$

$$+ \frac{1}{2} \sum_{u \in \bar{\mathcal{U}}(r)} \left| \int_{B_r(y^*)} \left( \check{\pi}_n(u \mid y) - \tilde{\pi}(u \mid y) \right) \mu_y(\mathrm{d}y) \right|,$$

it follows then that for all $y \in B_r(y^*)$ by (18) and (20), i.o. in $n$

$$\frac{1}{2} \sum_{u \in \bar{\mathcal{U}}(r)} \left| \int_{B_r(y^*)} (\check{\pi}_n(u \mid y) - \tilde{\pi}(u \mid y)) \, \mu_y(\mathrm{d}y) \right|$$

$$\geq \frac{1}{2} \sum_{u \in \bar{\mathcal{U}}(r)} \left| \int_{B_r(y^*)} (\check{\pi}_n(u \mid y^*) - \tilde{\pi}(u \mid y)) \, \mu_y(\mathrm{d}y) \right|$$

$$- \frac{1}{2} \sum_{u \in \bar{\mathcal{U}}(r)} \left| \int_{B_r(y^*)} (\check{\pi}_n(u \mid y^*) - \check{\pi}_n(u \mid y)) \, \mu_y(\mathrm{d}y) \right|$$

$$> \epsilon' - \frac{\epsilon'}{2} = \frac{\epsilon'}{2}. \tag{22}$$

**Pass to limit of Lipschitz functions.** Now consider the sequence of $m$-Lipschitz functions $f_m \colon \mathcal{Y} \times \mathcal{U} \to [0, 1]$,

$$f_m(y, u) = \min \left\{ 1, \left( 1 - (md(y, y^*) - mr + 1)^+ \right)^+ \right\}$$

$$\cdot \operatorname{sgn} \left( \int_{B_r(y^*)} (\check{\pi}_n(u \mid y^*) - \tilde{\pi}(u \mid y')) \, \mu_y(\mathrm{d}y') \right),$$

where $(\cdot)^+ = \max(\cdot, 0)$ and $\operatorname{sgn}$ is the sign function. Note that $f_m = 0$ for all $y \notin B_r(y^*)$. Further, as $m \to \infty$,

$$f_m(y, u) \uparrow \mathbf{1}_{B_r(y^*)}(y) \operatorname{sgn} \left( \int_{B_r(y^*)} (\check{\pi}_n(u \mid y^*) - \tilde{\pi}(u \mid y')) \, \mu_y(\mathrm{d}y') \right).$$

Then, by the prequel, we have by monotone convergence, as $m \to \infty$,

$$\iint f_m(y, u)(\check{\pi}_n(\mathrm{d}u \mid y) - \tilde{\pi}(\mathrm{d}u \mid y))\mu_y(\mathrm{d}y)$$

$$= \int_{B_r(y^*)} \sum_{u \in \mathcal{U}} f_m(y, u)(\check{\pi}_n(u \mid y) - \tilde{\pi}(u \mid y))\mu_y(\mathrm{d}y)$$

$$\to \int_{B_r(y^*)} \sum_{u \in \mathcal{U}} \operatorname{sgn} \left( \int_{B_r(y^*)} (\check{\pi}_n(u \mid y^*) - \tilde{\pi}(u \mid y')) \, \mu_y(\mathrm{d}y') \right) (\check{\pi}_n(u \mid y) - \tilde{\pi}(u \mid y))\mu_y(\mathrm{d}y)$$

$$= \sum_{u \in \bar{\mathcal{U}}(r)} \left| \int_{B_r(y^*)} (\check{\pi}_n(u \mid y) - \tilde{\pi}(u \mid y)) \, \mu_y(\mathrm{d}y) \right|$$

$$+ \sum_{u \notin \bar{\mathcal{U}}(r)} \operatorname{sgn} \left( \int_{B_r(y^*)} (\check{\pi}_n(u \mid y^*) - \tilde{\pi}(u \mid y')) \, \mu_y(\mathrm{d}y') \right) \int_{B_r(y^*)} (\check{\pi}_n(u \mid y) - \tilde{\pi}(u \mid y))\mu_y(\mathrm{d}y)$$

$$\geq \sum_{u \in \bar{\mathcal{U}}(r)} \left| \int_{B_r(y^*)} (\check{\pi}_n(u \mid y) - \tilde{\pi}(u \mid y)) \, \mu_y(\mathrm{d}y) \right|$$

$$- \sum_{u \notin \bar{\mathcal{U}}(r)} \left| \int_{B_r(y^*)} (\check{\pi}_n(u \mid y) - \check{\pi}_n(u \mid y^*))\mu_y(\mathrm{d}y) \right|$$

$$- \sum_{u \notin \bar{\mathcal{U}}(r)} \left| \int_{B_r(y^*)} (\check{\pi}_n(u \mid y^*) - \tilde{\pi}(u \mid y))\mu_y(\mathrm{d}y) \right|$$

$$> \frac{\epsilon'}{2} \cdot 2\mu_y(B_r(y^*)) - \frac{2\epsilon' - \epsilon}{4} \cdot 2\mu_y(B_r(y^*)) - \frac{\epsilon - \epsilon'}{2} \cdot 2\mu_y(B_r(y^*))$$

$$= \frac{1}{2}\left(\epsilon' - \frac{\epsilon}{2}\right)\mu_y(B_r(y^*)) > 0$$

i.o. in $n$, for the first term by (21) and (22), second by (20) and third by (19), noting that $\epsilon' > \frac{\epsilon}{2}$.

Hence, we may choose $m^*$ such that e.g.

$$\iint f_{m^*}(y, u)(\check{\pi}_n(\mathrm{d}u \mid y) - \tilde{\pi}(\mathrm{d}u \mid y))\mu_y(\mathrm{d}y) > \frac{1}{4}\left(\epsilon' - \frac{\epsilon}{2}\right)\mu_y(B_r(y^*)).$$

**Lower bound.** Finally, by noting that $\frac{1}{m^*}f_{m^*} \in \mathrm{Lip}(1)$ and applying the Kantorovich-Rubinstein duality, we have

$$
\begin{aligned}
W_1(\mu_y \otimes \check{\pi}_n, \mu_y \otimes \tilde{\pi}) &= \sup_{f \in \mathrm{Lip}(1)} \iint f(y, u)(\check{\pi}_n(\mathrm{d}u \mid y) - \tilde{\pi}(\mathrm{d}u \mid y))\mu_y(\mathrm{d}y) \\
&\geq \iint \frac{1}{m^*}f_{m^*}(y, u)(\check{\pi}_n(\mathrm{d}u \mid y) - \tilde{\pi}(\mathrm{d}u \mid y))\mu_y(\mathrm{d}y) \\
&> \frac{1}{m^*}\frac{1}{4}\left(\epsilon' - \frac{\epsilon}{2}\right)\mu_y(B_r(y^*)) > 0
\end{aligned}
$$

i.o. in $n$, and therefore $\mu_y \otimes \check{\pi}_n \not\to \mu_y \otimes \tilde{\pi}$. But $\tilde{h} = \mu_y \otimes \tilde{\pi}$ was assumed to be the limit of $\mu_y \otimes \check{\pi}_n$, leading to a contradiction. Hence, $\mu_y$-a.e. convergence must hold.

**(ii) Lipschitz extension of lower-level policies.** For finite actions, note that $\mathcal{P}(\mathcal{U})$ is (Lipschitz) equivalent to a subset of the Hilbert space $\mathbb{R}^{|\mathcal{U}|}$. Therefore, by the Kirszbraun-Valentine theorem (see e.g. (Cobzaş et al., 2019, Theorem 4.2.3)), we can modify $\tilde{\pi}$ to be $L_\Pi$-Lipschitz not only $\mu_y$-a.e., but on full $\mathcal{Y}$.

**(iii) Equality of limits.** We show that for any $\epsilon > 0$, $W_1(h, \mu_y \otimes \tilde{\pi}) < \epsilon$, which implies $W_1(h, \mu_y \otimes \tilde{\pi}) = 0$ and therefore $h = \mu_y \otimes \tilde{\pi}$. First, note that by the triangle inequality, we have

$$W_1(h, \mu_y \otimes \tilde{\pi}) \leq W_1(h, \mu_y \otimes \check{\pi}_n) + W_1(\mu_y \otimes \check{\pi}_n, \mu_y \otimes \tilde{\pi})$$

and thus by $\mu_y \otimes \check{\pi}_n \to h$ for sufficiently large $n$, it suffices to show $W_1(\mu_y \otimes \check{\pi}_n, \mu_y \otimes \tilde{\pi}) < \epsilon$.

By the prequel, we choose a measurable set $A \subseteq \mathcal{Y}$ such that $\mu_y(A) < \frac{\epsilon}{2\,\mathrm{diam}(\mathcal{U})}$ and $\check{\pi}_n(y)$ converges uniformly on $\mathcal{Y} \setminus A$. Now by uniform convergence, we choose $n$ sufficiently large such that $W_1(\check{\pi}_n(y), \tilde{\pi}(y)) < \frac{\epsilon}{2}$ on $\mathcal{Y} \setminus A$. By Kantorovich-Rubinstein duality, we have

$$
\begin{aligned}
W_1(\mu_y \otimes \check{\pi}_n, \mu_y \otimes \tilde{\pi}) &= \sup_{f \in \mathrm{Lip}(1)} \iint f(y, u)(\check{\pi}_n(\mathrm{d}u \mid y) - \tilde{\pi}(\mathrm{d}u \mid y))\mu_y(\mathrm{d}y) \\
&\leq \int \left(\sup_{f \in \mathrm{Lip}(1)} \int f(y, u)(\check{\pi}_n(\mathrm{d}u \mid y) - \tilde{\pi}(\mathrm{d}u \mid y))\right)\mu_y(\mathrm{d}y) \\
&= \int W_1(\check{\pi}_n(y), \tilde{\pi}(y))\mu_y(\mathrm{d}y) \\
&= \int_A W_1(\check{\pi}_n(y), \tilde{\pi}(y))\mu_y(\mathrm{d}y) + \int_{\mathcal{Y} \setminus A} W_1(\check{\pi}_n(y), \tilde{\pi}(y))\mu_y(\mathrm{d}y) \\
&< \frac{\epsilon}{2\,\mathrm{diam}(\mathcal{U})}\,\mathrm{diam}(\mathcal{U}) + \left(1 - \frac{\epsilon}{2\,\mathrm{diam}(\mathcal{U})}\right)\frac{\epsilon}{2} < \epsilon.
\end{aligned}
$$

This completes the proof. $\qquad\square$

## L  EQUIVALENCE OF DEC-MFC AND DEC-MFC MDP

*Proof of Proposition 2.* The proof is similar to the proof of Proposition 1 by induction. We begin by showing the first statement. We show $\bar{\mu}_t = \hat{\mu}_t$ at all times $t \in \mathcal{T}$, as it then follows that

$\bar{J}(\bar{\pi}) = \sum_{t=0}^{\infty} \gamma^t r(\bar{\mu}_t) = \sum_{t=0}^{\infty} \gamma^t r(\hat{\mu}_t) = \hat{J}(\hat{\pi})$ under deterministic $\hat{\pi} \in \hat{\Pi}$. At time $t = 0$, we have by definition $\bar{\mu}_0 = \mu_0 = \hat{\mu}_0$. Assume $\bar{\mu}_t = \hat{\mu}_t$ at time $t$, then at time $t + 1$, we have

$$\hat{\mu}_{t+1} = \hat{T}(\hat{\mu}_t, h_t) = \iiint P(x, u, \hat{\mu}_t)\check{\pi}[h_t](\mathrm{d}u \mid y)P^y(\mathrm{d}y \mid x, \hat{\mu}_t)\hat{\mu}_t(\mathrm{d}x) \tag{23}$$

$$= \iiint P(x, u, \mu_t)\bar{\pi}_t(\mathrm{d}u \mid y, \bar{\mu}_t)P^y(\mathrm{d}y \mid x, \bar{\mu}_t)\bar{\mu}_t(\mathrm{d}x) = \bar{\mu}_{t+1} \tag{24}$$

by definition of $\bar{\pi}_t(\nu) = \check{\pi}[h_t]$, which is the desired statement. An analogous proof for the second statement in the opposite direction completes the proof. $\qquad\square$

## M   OPTIMALITY OF DEC-MFC MDP SOLUTIONS

*Proof of Corollary 3.* As in the proof of Corollary 2, we first show Dec-POMFC optimality of $\Phi(\Psi(\hat{\pi}))$. Assume $J(\Phi(\Psi(\hat{\pi}))) < \sup_{\pi' \in \Pi} J(\pi')$. Then there exists $\pi' \in \Pi$ such that $J(\Phi(\Psi(\hat{\pi}))) < J(\pi')$. But by Proposition 1, there exists $\bar{\pi}' \in \bar{\Pi}$ such that $\bar{J}(\bar{\pi}') = J(\pi')$. Further, by Proposition 2, there exists $\hat{\pi}' \in \bar{\Pi}$ such that $\bar{J}(\bar{\pi}') = \hat{J}(\hat{\pi}')$. Thus, $\hat{J}(\hat{\pi}) = \bar{J}(\bar{\pi}) = J(\Phi(\Psi(\hat{\pi}))) < J(\pi') = \bar{J}(\bar{\pi}') = \hat{J}(\hat{\pi}')$, which contradicts $\hat{\pi} \in \arg\max_{\hat{\pi}'} \hat{J}(\hat{\pi}')$. Therefore, $\Phi(\Psi(\hat{\pi})) \in \arg\max_{\pi' \in \Pi} J(\pi')$. Hence, $\Phi(\Psi(\hat{\pi}))$ fulfills the conditions of Corollary 1, completing the proof. $\qquad\square$

## N   LIPSCHITZ CONTINUITY OF RBF KERNELS

*Proof of Proposition 3.* First, note that

$$|\nabla_y \kappa(y_b, y)| = \exp\left(\frac{-\|y_b - y\|^2}{2\sigma^2}\right)\frac{|\langle y_b - y, y\rangle|}{2\sigma^2}$$

$$\leq \frac{1}{2\sigma^2}\operatorname{diam}(\mathcal{Y})\max_{y \in \mathcal{Y}}\|y\|$$

for diameter $\operatorname{diam}(\mathcal{Y}) < \infty$ by compactness of $\mathcal{Y}$, which is equal one for discrete spaces. Further,

$$\left|\sum_{b' \in [M_{\mathcal{Y}}]} \kappa(y_{b'}, y)\right| = \sum_{b' \in [M_{\mathcal{Y}}]} \kappa(y_{b'}, y) \geq M_{\mathcal{Y}}\exp\left(-\frac{\operatorname{diam}(\mathcal{Y})^2}{2\sigma^2}\right)$$

and $|\kappa(y_b, y)| \leq 1$.

Hence, the RBF kernel $y \mapsto \kappa(y_b, y)p_b = \exp(\frac{-\|y_b - y\|^2}{2\sigma^2})$ with parameter $\sigma^2 > 0$ on $\mathcal{Y}$ is Lipschitz for any $b \in [M_{\mathcal{Y}}]$, since for any $y, y' \in \mathcal{Y}$,

$$\left|\nabla_y\left(Z^{-1}(y)\kappa(y_b, y)\right)\right|$$

$$= \left|\frac{\nabla_y\kappa(y_b, y)\sum_{b' \in [M_{\mathcal{Y}}]}\kappa(y_{b'}, y) + \sum_{b' \in [M_{\mathcal{Y}}]}\nabla_y\kappa(y_{b'}, y)\kappa(y_b, y)}{\left(\sum_{b' \in [M_{\mathcal{Y}}]}\kappa(y_{b'}, y)\right)^2}\right|$$

$$\leq \frac{1}{M_{\mathcal{Y}}^2\exp^2\left(-\frac{\operatorname{diam}(\mathcal{Y})^2}{2\sigma^2}\right)}\left(\frac{1}{2\sigma^2}\operatorname{diam}(\mathcal{Y})\max_{y \in \mathcal{Y}}\|y\|M_{\mathcal{Y}} + M_{\mathcal{Y}}\frac{1}{2\sigma^2}\operatorname{diam}(\mathcal{Y})\max_{y \in \mathcal{Y}}\|y\|\right)$$

$$= \frac{\operatorname{diam}(\mathcal{Y})\max_{y \in \mathcal{Y}}\|y\|}{\sigma^2 M_{\mathcal{Y}}\exp^2\left(-\frac{\operatorname{diam}(\mathcal{Y})^2}{2\sigma^2}\right)}$$

for any $b \in [M_{\mathcal{Y}}]$. Hence, by noting that the following supremum is invariant to addition of constants,

$$W_1\left(Z^{-1}(y)\sum_{b \in [M_{\mathcal{Y}}]}\kappa(y_b, y)p_b, Z^{-1}(y')\sum_{b \in [M_{\mathcal{Y}}]}\kappa(y_b, y')p_b\right)$$

$$= \sup_{f \in \mathrm{Lip}(1)} \int f \left( Z^{-1}(y) \sum_{b \in [M_{\mathcal{Y}}]} \kappa(y_b, y) \mathrm{d}p_b - Z^{-1}(y') \sum_{b \in [M_{\mathcal{Y}}]} \kappa(y_b, y') \mathrm{d}p_b \right)$$

$$= \sup_{f \in \mathrm{Lip}(1), |f| \le \frac{1}{2} \mathrm{diam}(\mathcal{U})} \int f \left( Z^{-1}(y) \sum_{b \in [M_{\mathcal{Y}}]} \kappa(y_b, y) - Z^{-1}(y') \sum_{b \in [M_{\mathcal{Y}}]} \kappa(y_b, y') \right) \mathrm{d}p_b$$

$$\le \sum_{b \in [M_{\mathcal{Y}}]} \left| Z^{-1}(y)\kappa(y_b, y) - Z^{-1}(y')\kappa(y_b, y') \right| \sup_{f \in \mathrm{Lip}(1), |f| \le \frac{1}{2} \mathrm{diam}(\mathcal{U})} \int f \mathrm{d}p_b$$

$$\le M_{\mathcal{Y}} \frac{\mathrm{diam}(\mathcal{Y}) \max_{y \in \mathcal{Y}} \|y\|}{\sigma^2 M_{\mathcal{Y}} \exp^2 \left( -\frac{1}{2\sigma^2} \mathrm{diam}(\mathcal{Y})^2 \right)} \|y - y'\| \cdot \frac{1}{2} \mathrm{diam}(\mathcal{U}).$$

which is $L_\Pi$-Lipschitz if

$$\frac{\mathrm{diam}(\mathcal{Y}) \mathrm{diam}(\mathcal{U}) \max_{y \in \mathcal{Y}} \|y\|}{2\sigma^2 \exp^2 \left( -\frac{1}{2\sigma^2} \mathrm{diam}(\mathcal{Y})^2 \right)} \le L_\Pi$$

$$\iff \sigma^2 \exp^2 \left( -\frac{1}{2\sigma^2} \mathrm{diam}(\mathcal{Y})^2 \right) \ge \frac{1}{L_\Pi} \mathrm{diam}(\mathcal{Y}) \mathrm{diam}(\mathcal{U}) \max_{y \in \mathcal{Y}} \|y\|.$$

Note that such $\sigma^2 > 0$ exists, as $\sigma^2 \exp^2 \left( -\frac{1}{2\sigma^2} \mathrm{diam}(\mathcal{Y})^2 \right) \to +\infty$ as $\sigma^2 \to +\infty$. $\qquad \square$

## O  POLICY GRADIENT APPROXIMATION

*Proof of Theorem 3.* Keeping in mind that we have the **centralized training** system for stationary policy $\hat{\pi}^\theta$ parametrized by $\theta$,

$$\tilde{\xi}_t \sim \hat{\pi}^\theta(\tilde{\mu}_t^N), \quad \check{\pi}_t = \Lambda(\tilde{\xi}_t)$$

$$\tilde{y}_t^i \sim P^y(\tilde{y}_t^i \mid \tilde{x}_t^i, \tilde{\mu}_t^N), \quad \tilde{u}_t^i \sim \check{\pi}_t(\tilde{u}_t^i \mid \tilde{y}_t^i), \quad \tilde{x}_{t+1}^i \sim P(\tilde{x}_{t+1}^i \mid \tilde{x}_t^i, \tilde{u}_t^i, \tilde{\mu}_t^N), \quad \forall i \in [N],$$

which we obtained by parametrizing the MDP actions via parametrizations $\xi \in \Xi$, the equivalent Dec-MFC MDP system concomitant with (4) under parametrization $\Lambda(\xi)$ for lower-level policies is

$$\xi_t \sim \hat{\pi}^\theta(\hat{\mu}_t), \quad \hat{\mu}_{t+1} = \hat{T}(\hat{\mu}_t, \xi_t) \coloneqq \iiint P(x, u, \hat{\mu}_t)\Lambda(\xi_t)(\mathrm{d}u \mid y)P^y(\mathrm{d}y \mid x, \hat{\mu}_t)\hat{\mu}_t(\mathrm{d}x) \quad (25)$$

where we now sample $\xi_t$ instead of $h_t$. Note that for kernel representations, this new $\hat{T}$ is indeed Lipschitz, which follows from Lipschitzness of $\hat{\mu}_t \otimes P^y(\hat{\mu}_t) \otimes \Lambda(\xi_t)$ in $(\hat{\mu}_t, \xi_t)$.

**Lemma 4.** *Under Assumptions 1a and 3, the transitions $\hat{T}$ of the system with parametrized actions are $L_{\hat{T}}$-Lipschitz with $L_{\hat{T}} \coloneqq 2L_P + L_P L_\lambda + 2L' L_{P^y}$.*

Proofs for lemmas are found in their respective following sections.

First, we prove $d_{\hat{\pi}^\theta}^N \to d_{\hat{\pi}^\theta}$ in $\mathcal{P}(\mathcal{P}(\mathcal{X}))$ by showing at any time $t$ that under $\hat{\pi}^\theta$, the centralized training system MF $\tilde{\mu}_t^N$ converges to the limiting Dec-MFC MF $\hat{\mu}_t$ in (25). The convergence is in the same sense as in Theorem 1.

**Lemma 5.** *For any equicontinuous family of functions $\mathcal{F} \subseteq \mathbb{R}^{\mathcal{P}(\mathcal{X})}$, under Assumptions 1a–1b and 3, at all times $t$ we have*

$$\sup_{f \in \mathcal{F}} \left| \mathbb{E} \left[ f(\tilde{\mu}_t^N) - f(\hat{\mu}_t) \right] \right| \to 0. \quad (26)$$

We also show that $\tilde{Q}^\theta(\mu, \xi) \to Q^\theta(\mu, \xi)$, since we can show the same convergence as in (26) for new conditional systems, where for any $\mu, \xi$ we let $\tilde{\mu}_0 = \mu = \mu_0$ and $\tilde{\xi}_0 = \xi = \xi_0$ at time zero, where $\tilde{\mu}_0$ is the initial state distribution of the centralized training system.

**Lemma 6.** *Under Assumptions 1a and 3, as $N \to \infty$, we have for any $\mu \in \mathcal{P}(\mathcal{X})$, $\xi \in \Xi$ that*

$$\left| \tilde{Q}^\theta(\mu, \xi) - Q^\theta(\mu, \xi) \right| \to 0.$$

Furthermore, $Q^\theta(\mu, \xi)$ is also continuous by a similar argument.

**Lemma 7.** *For any equicontinuous family of functions $\mathcal{F} \subseteq \mathbb{R}^{\mathcal{P}(\mathcal{X})}$, under Assumptions 1a and 3, at all times $t \in \mathcal{T}$, the conditional expectations version of the MF is continuous in the starting conditions, in the sense that for any $(\mu_n, \xi_n) \to (\mu, \xi)$,*

$$\sup_{f \in \mathcal{F}} |\mathbb{E}\left[f(\hat{\mu}_t) \mid \hat{\mu}_0 = \mu_n, \xi_0 = \xi_n\right] - \mathbb{E}\left[f(\hat{\mu}_t) \mid \hat{\mu}_0 = \mu, \xi_0 = \xi\right]| \to 0.$$

Lastly, keeping in mind $d_{\hat{\pi}^\theta}^N = (1 - \gamma) \sum_{t \in \mathcal{T}} \gamma^t \mathcal{L}_{\hat{\pi}^\theta}(\tilde{\mu}_t^N)$, we have the desired statement

$$\left\| (1-\gamma)^{-1} \mathbb{E}_{\mu \sim d_{\hat{\pi}^\theta}^N, \xi \sim \hat{\pi}^\theta(\mu)} \left[ \tilde{Q}^\theta(\mu, \xi) \nabla_\theta \log \hat{\pi}^\theta(\xi \mid \mu) \right] - \nabla_\theta J(\hat{\pi}^\theta) \right\|$$

$$\leq (1-\gamma)^{-1} \left\| \mathbb{E}_{\mu \sim d_{\hat{\pi}^\theta}^N, \xi \sim \hat{\pi}^\theta(\mu)} \left[ \left( \tilde{Q}^\theta(\mu, \xi) - Q^\theta(\mu, \xi) \right) \nabla_\theta \log \hat{\pi}^\theta(\xi \mid \mu) \right] \right\|$$

$$+ \left\| (1-\gamma)^{-1} \mathbb{E}_{\mu \sim d_{\hat{\pi}^\theta}^N, \xi \sim \hat{\pi}^\theta(\mu)} \left[ Q^\theta(\mu, \xi) \nabla_\theta \log \hat{\pi}^\theta(\xi \mid \mu) \right] - \nabla_\theta J(\hat{\pi}^\theta) \right\|$$

$$\leq (1-\gamma)^{-1} \left\| \mathbb{E}_{\mu \sim d_{\hat{\pi}^\theta}^N, \xi \sim \hat{\pi}^\theta(\mu)} \left[ \left( \tilde{Q}^\theta(\mu, \xi) - Q^\theta(\mu, \xi) \right) \nabla_\theta \log \hat{\pi}^\theta(\xi \mid \mu) \right] \right\|$$

$$+ \left\| \sum_{t=T}^{\infty} \gamma^t \mathbb{E}_{\xi \sim \hat{\pi}^\theta(\tilde{\mu}_t^N)} \left[ Q^\theta(\tilde{\mu}_t^N, \xi) \nabla_\theta \log \hat{\pi}^\theta(\xi \mid \tilde{\mu}_t^N) - Q^\theta(\hat{\mu}_t, \xi) \nabla_\theta \log \hat{\pi}^\theta(\xi \mid \hat{\mu}_t) \right] \right\|$$

$$+ \left\| \sum_{t=0}^{T-1} \gamma^t \mathbb{E}_{\xi \sim \hat{\pi}^\theta(\tilde{\mu}_t^N)} \left[ Q^\theta(\tilde{\mu}_t^N, \xi) \nabla_\theta \log \hat{\pi}^\theta(\xi \mid \tilde{\mu}_t^N) - Q^\theta(\hat{\mu}_t, \xi) \nabla_\theta \log \hat{\pi}^\theta(\xi \mid \hat{\mu}_t) \right] \right\|$$

$$\to 0$$

for the **first** term from $\tilde{Q}^\theta(\mu, \xi) \to Q^\theta(\mu, \xi)$ uniformly by Lemma 6 and compactness of the domain, for the **second** by Assumption 3 and 1a uniformly bounding $\nabla_\theta \log \pi^\theta$, $Q^\theta$ and choosing sufficiently large $T$, and for the **third** by repeating the argument for $Q$: Notice that

$$\left\| \sum_{t=0}^{T-1} \gamma^t \mathbb{E}_{\xi \sim \hat{\pi}^\theta(\tilde{\mu}_t^N)} \left[ Q^\theta(\tilde{\mu}_t^N, \xi) \nabla_\theta \log \hat{\pi}^\theta(\xi \mid \tilde{\mu}_t^N) - Q^\theta(\hat{\mu}_t, \xi) \nabla_\theta \log \hat{\pi}^\theta(\xi \mid \hat{\mu}_t) \right] \right\|$$

$$\leq \left\| \sum_{t=0}^{T-1} \gamma^t \mathbb{E}_{\xi \sim \hat{\pi}^\theta(\tilde{\mu}_t^N)} \left[ \sum_{t'=T'}^{\infty} \gamma^{t'} \left( \mathbb{E}\left[r(\hat{\mu}_{t'}) \mid \hat{\mu}_0 = \tilde{\mu}_t^N, \xi_0 = \xi\right] \nabla_\theta \log \hat{\pi}^\theta(\xi \mid \tilde{\mu}_t^N) \right. \right. \right.$$

$$\left. \left. \left. - \mathbb{E}\left[r(\hat{\mu}_{t'}) \mid \hat{\mu}_0 = \hat{\mu}_t, \xi_0 = \xi\right] \nabla_\theta \log \hat{\pi}^\theta(\xi \mid \hat{\mu}_t) \right) \right] \right\|$$

$$+ \left\| \sum_{t=0}^{T-1} \gamma^t \mathbb{E}_{\xi \sim \hat{\pi}^\theta(\tilde{\mu}_t^N)} \left[ \sum_{t'=0}^{T'-1} \gamma^{t'} \left( \mathbb{E}\left[r(\hat{\mu}_{t'}) \mid \hat{\mu}_0 = \tilde{\mu}_t^N, \xi_0 = \xi\right] \nabla_\theta \log \hat{\pi}^\theta(\xi \mid \tilde{\mu}_t^N) \right. \right. \right.$$

$$\left. \left. \left. - \mathbb{E}\left[r(\hat{\mu}_{t'}) \mid \hat{\mu}_0 = \hat{\mu}_t, \xi_0 = \xi\right] \nabla_\theta \log \hat{\pi}^\theta(\xi \mid \hat{\mu}_t) \right) \right] \right\|,$$

where the inner expectations are on the conditional system. Letting $T'$ sufficiently large bounds the **former** term by uniform bounds on the summands from Assumption 3. Then, for the **latter** term, apply Lemma 5 at times $t' < T'$ to the functions $f(\mu) = \int \mathbb{E}\left[r(\hat{\mu}_{t'}) \mid \hat{\mu}_0 = \mu, \xi_0 = \xi\right] \nabla_\theta \log \hat{\pi}^\theta(\xi \mid \mu) \hat{\pi}^\theta(\mathrm{d}\xi \mid \mu) \mathrm{d}\xi$, which are continuous up to any finite time $t'$ by Lemma 7 and Assumption 3. $\qquad\square$

# P  LIPSCHITZ CONTINUITY OF TRANSITIONS UNDER PARAMETRIZED ACTIONS

*Proof of Lemma 4.* We have by definition

$$\iiint P(x, u, \hat{\mu}) \Lambda(\xi)(\mathrm{d}u \mid y) P^y(\mathrm{d}y \mid x, \hat{\mu}) \hat{\mu}(\mathrm{d}x)$$

$$= \iiint P(x, u, \hat{\mu}) \frac{\sum_{b \in [M_\mathcal{Y}]} \kappa(y_b, y) \lambda_b(\xi)(\mathrm{d}u)}{\sum_{b \in [M_\mathcal{Y}]} \kappa(y_b, y)} P^y(\mathrm{d}y \mid x, \hat{\mu}) \hat{\mu}(\mathrm{d}x).$$

Consider any $\xi, \xi' \in \Xi$, $\hat{\mu}, \hat{\mu}' \in \mathcal{P}(\mathcal{X})$. Then, for readability, write

$$\hat{\mu}_{xy} := \hat{\mu} \otimes P^y(\hat{\mu}), \quad \hat{\mu}_{xyu} := \hat{\mu}_{xy} \otimes \Lambda(\xi), \quad \hat{\mu}_{xyux'} := \hat{\mu}_{xyu} \otimes P(\hat{\mu}),$$
$$\hat{\mu}'_{xy} := \hat{\mu}' \otimes P^y(\hat{\mu}'), \quad \hat{\mu}'_{xyu} := \hat{\mu}'_{xy} \otimes \Lambda(\xi'), \quad \hat{\mu}'_{xyux'} := \hat{\mu}'_{xyu} \otimes P(\hat{\mu}'),$$
$$\Delta P(\cdot \mid x, u) := P(\cdot \mid x, u, \hat{\mu}) - P(\cdot \mid x, u, \hat{\mu}'),$$
$$\Delta \Lambda(\cdot \mid y) := \frac{\sum_b \kappa(y_b, y) \left( \lambda_b(\xi)(\cdot) - \lambda_b(\xi')(\cdot) \right)}{\sum_b \kappa(y_b, y)},$$
$$\Delta P^y(\cdot \mid x) := P^y(\cdot \mid x, \hat{\mu}) - P^y(\cdot \mid x, \hat{\mu}'), \quad \Delta \mu := \hat{\mu} - \hat{\mu}'$$

to obtain

$$W_1 \left( \iiint P(x, u, \hat{\mu}) \frac{\sum_b \kappa(y_b, y) \lambda_b(\xi)(\mathrm{d}u)}{\sum_b \kappa(y_b, y)} P^y(\mathrm{d}y \mid x, \hat{\mu}) \hat{\mu}(\mathrm{d}x), \right.$$
$$\left. \iiint P(x, u, \hat{\mu}') \frac{\sum_b \kappa(y_b, y) \lambda_b(\xi')(\mathrm{d}u)}{\sum_b \kappa(y_b, y)} P^y(\mathrm{d}y \mid x, \hat{\mu}') \hat{\mu}'(\mathrm{d}x) \right)$$

$$= \sup_{f \in \mathrm{Lip}(1)} \iiiint f(x') \left( \hat{\mu}_{xyux'}(\mathrm{d}x, \mathrm{d}y, \mathrm{d}u, \mathrm{d}x') - \hat{\mu}'_{xyux'}(\mathrm{d}x, \mathrm{d}y, \mathrm{d}u, \mathrm{d}x') \right)$$

$$\leq \sup_{f \in \mathrm{Lip}(1)} \iiiint f(x') \Delta P(\mathrm{d}x' \mid x, u) \hat{\mu}_{xyu}(\mathrm{d}x, \mathrm{d}y, \mathrm{d}u)$$

$$+ \sup_{f \in \mathrm{Lip}(1)} \iiiint f(x') P(\mathrm{d}x' \mid x, u, \hat{\mu}') \Delta \Lambda(\mathrm{d}u \mid y) \hat{\mu}_{xy}(\mathrm{d}x, \mathrm{d}y))$$

$$+ \sup_{f \in \mathrm{Lip}(1)} \iiiint f(x') P(\mathrm{d}x' \mid x, u, \hat{\mu}') \frac{\sum_b \kappa(y_b, y) \lambda_b(\xi')(\mathrm{d}u)}{\sum_b \kappa(y_b, y)} \Delta P^y(\mathrm{d}y \mid x) \hat{\mu}(\mathrm{d}x)$$

$$+ \sup_{f \in \mathrm{Lip}(1)} \iiiint f(x') P(\mathrm{d}x' \mid x, u, \hat{\mu}') \frac{\sum_b \kappa(y_b, y) \lambda_b(\xi')(\mathrm{d}u)}{\sum_b \kappa(y_b, y)} P^y(\mathrm{d}y \mid x, \hat{\mu}') \Delta \mu(\mathrm{d}x)$$

$$\leq \sup_{f \in \mathrm{Lip}(1)} \sup_{(x,y,u) \in \mathcal{X} \times \mathcal{Y} \times \mathcal{U}} \left| \int f(x') \Delta P(\mathrm{d}x' \mid x, u) \right|$$

$$+ \sup_{f \in \mathrm{Lip}(1)} \sup_{(x,y) \in \mathcal{X} \times \mathcal{Y}} \left| \iint f(x') P(\mathrm{d}x' \mid x, u, \hat{\mu}') \Delta \Lambda(\mathrm{d}u \mid y) \right|$$

$$+ \sup_{f \in \mathrm{Lip}(1)} \sup_{x \in \mathcal{X}} \left| \iiint f(x') P(\mathrm{d}x' \mid x, u, \hat{\mu}') \frac{\sum_b \kappa(y_b, y) \lambda_b(\xi')(\mathrm{d}u)}{\sum_b \kappa(y_b, y)} \Delta P^y(\mathrm{d}y \mid x) \right|$$

$$+ \sup_{f \in \mathrm{Lip}(1)} \left| \iiiint f(x') P(\mathrm{d}x' \mid x, u, \hat{\mu}') \frac{\sum_b \kappa(y_b, y) \lambda_b(\xi')(\mathrm{d}u)}{\sum_b \kappa(y_b, y)} P^y(\mathrm{d}y \mid x, \hat{\mu}') \Delta \mu(\mathrm{d}x) \right|$$

bounded by the same arguments as in Theorem 1:

For the **first** term, we have that the function $x' \mapsto f(x')$ is 1-Lipschitz, and therefore

$$\sup_{f \in \mathrm{Lip}(1)} \sup_{(x,y,u) \in \mathcal{X} \times \mathcal{Y} \times \mathcal{U}} \left| \int f(x') \left( P(\mathrm{d}x' \mid x, u, \hat{\mu}) - P(\mathrm{d}x' \mid x, u, \hat{\mu}') \right) \right| \leq L_P W_1(\hat{\mu}, \hat{\mu}')$$

by Assumption 1a.

For the **second** term, we have $L_P$-Lipschitz $u \mapsto \int f(x') P(\mathrm{d}x' \mid x, u, \hat{\mu}')$, since for any $f \in \mathrm{Lip}(1)$ and $(x, y) \in \mathcal{X} \times \mathcal{Y}$, we obtain

$$\left| \int f(x') P(\mathrm{d}x' \mid x, u, \hat{\mu}') - \int f(x') P(\mathrm{d}x' \mid x, u', \hat{\mu}') \right|$$
$$\leq W_1(P(x, u, \hat{\mu}'), P(x, u', \hat{\mu}')) \leq L_P d(u, u')$$

for any $u, u' \in \mathcal{U}$ by Assumption 1a, and therefore

$$\sup_{f \in \mathrm{Lip}(1)} \sup_{(x,y) \in \mathcal{X} \times \mathcal{Y}} \left| \iint f(x') P(\mathrm{d}x' \mid x, u, \hat{\mu}') \frac{\sum_b \kappa(y_b, y) \left( \lambda_b(\xi)(\mathrm{d}u) - \lambda_b(\xi')(\mathrm{d}u) \right)}{\sum_b \kappa(y_b, y)} \right|$$

$$\leq \frac{\sum_b \kappa(y_b, y) L_P W_1\left(\lambda_b(\xi), \lambda_b(\xi')\right)}{\sum_b \kappa(y_b, y)} \leq L_P L_\lambda d(\xi, \xi')$$

by Assumption 3.

For the **third** term, we have $L'$-Lipschitz $y \mapsto \iint f(x') P(\mathrm{d}x' \mid x, u, \hat{\mu}') \frac{\sum_b \kappa(y_b, y) \lambda_b(\xi')(\mathrm{d}u)}{\sum_b \kappa(y_b, y)}$ where we define $L' := L_P \frac{\mathrm{diam}(\mathcal{Y}) \, \mathrm{diam}(\mathcal{U}) \, \max_{y \in \mathcal{Y}} \|y\|}{2\sigma^2 \exp^2\left(-\frac{1}{2\sigma^2} \mathrm{diam}(\mathcal{Y})^2\right)}$, since for any $f \in \mathrm{Lip}(1)$ and $x \in \mathcal{X}$, we obtain

$$\left| \iint f(x') P(\mathrm{d}x' \mid x, u, \hat{\mu}') \left( \frac{\sum_b \kappa(y_b, y) \lambda_b(\xi')(\mathrm{d}u)}{\sum_b \kappa(y_b, y)} - \frac{\sum_b \kappa(y_b, y') \lambda_b(\xi')(\mathrm{d}u)}{\sum_b \kappa(y_b, y')} \right) \right|$$
$$\leq L_P \cdot \frac{\mathrm{diam}(\mathcal{Y}) \, \mathrm{diam}(\mathcal{U}) \, \max_{y \in \mathcal{Y}} \|y\|}{2\sigma^2 \exp^2\left(-\frac{1}{2\sigma^2} \mathrm{diam}(\mathcal{Y})^2\right)} d(y, y')$$

for any $y, y' \in \mathcal{Y}$ by Proposition 3 and the prequel, and therefore

$$\sup_{f \in \mathrm{Lip}(1), x} \left| \iiint f(x') P(\mathrm{d}x' \mid x, u, \hat{\mu}') \frac{\sum_b \kappa(y_b, y) \lambda_b(\xi')(\mathrm{d}u)}{\sum_b \kappa(y_b, y)} \left(P^y(\mathrm{d}y \mid x, \hat{\mu}) - P^y(\mathrm{d}y \mid x, \hat{\mu}')\right) \right|$$
$$\leq L' L_{P^y} W_1(\hat{\mu}, \hat{\mu}')$$

by Assumption 1a.

Lastly, for the **fourth** term, $x \mapsto \iiint f(x') P(\mathrm{d}x' \mid x, u, \hat{\mu}') \frac{\sum_b \kappa(y_b, y) \lambda_b(\xi')(\mathrm{d}u)}{\sum_b \kappa(y_b, y)} P^y(\mathrm{d}y \mid x, \hat{\mu}')$ is similarly $(L_P + L' L_{P^y})$-Lipschitz, since for any $f \in \mathrm{Lip}(1)$, we obtain

$$\left| \iiint f(x') P(\mathrm{d}x' \mid x, u, \hat{\mu}') \frac{\sum_b \kappa(y_b, y) \lambda_b(\xi')(\mathrm{d}u)}{\sum_b \kappa(y_b, y)} P^y(\mathrm{d}y \mid x, \hat{\mu}') \right.$$
$$\left. - \iiint f(x') P(\mathrm{d}x' \mid x'', u, \hat{\mu}') \frac{\sum_b \kappa(y_b, y) \lambda_b(\xi')(\mathrm{d}u)}{\sum_b \kappa(y_b, y)} P^y(\mathrm{d}y \mid x'', \hat{\mu}') \right|$$
$$\leq \left| \iiint f(x') \left(P(\mathrm{d}x' \mid x, u, \hat{\mu}') - P(\mathrm{d}x' \mid x'', u, \hat{\mu}')\right) \frac{\sum_b \kappa(y_b, y) \lambda_b(\xi')(\mathrm{d}u)}{\sum_b \kappa(y_b, y)} P^y(\mathrm{d}y \mid x, \hat{\mu}') \right|$$
$$+ \left| \iiint f(x') P(\mathrm{d}x' \mid x'', u, \hat{\mu}') \frac{\sum_b \kappa(y_b, y) \lambda_b(\xi')(\mathrm{d}u)}{\sum_b \kappa(y_b, y)} \left(P^y(\mathrm{d}y \mid x, \hat{\mu}') - P^y(\mathrm{d}y \mid x'', \hat{\mu}')\right) \right|$$
$$\leq L_P d(x, x'') + L' L_{P^y} d(x, x'') = (L_P + L' L_{P^y}) d(x, x'')$$

for any $x, x'' \in \mathcal{X}$ by the prequel, which implies

$$\sup_{f \in \mathrm{Lip}(1)} \left| \iiiint f(x') P(\mathrm{d}x' \mid x, u, \hat{\mu}') \frac{\sum_b \kappa(y_b, y) \lambda_b(\xi')(\mathrm{d}u)}{\sum_b \kappa(y_b, y)} P^y(\mathrm{d}y \mid x, \hat{\mu}') \left(\hat{\mu}(\mathrm{d}x) - \hat{\mu}'(\mathrm{d}x)\right) \right|$$
$$\leq (L_P + L' L_{P^y}) W_1(\hat{\mu}, \hat{\mu}').$$

Overall, the map $\hat{T}$ is therefore Lipschitz with constant $L_{\hat{T}} = 2L_P + L_P L_\lambda + 2L' L_{P^y}$. $\qquad \square$

## Q (CENTRALIZED) PROPAGATION OF CHAOS

*Proof of Lemma 5.* The proof is the same as the proof of Theorem 1. The only difference is that for the weak LLN argument, we condition not only on $\tilde{x}_t^N$, but also on $\tilde{\xi}_t$, while for the induction assumption, we still apply to equicontinuous functions by Assumption 3 and Lemma 4.

In other words, for the weak LLN we use

$$\mathbb{E}\left[d_\Sigma\left(\tilde{\mu}_{t+1}^N, \tilde{T}\left(\tilde{\mu}_t^N, \tilde{\xi}_t\right)\right)\right] = \sum_{m=1}^\infty 2^{-m} \mathbb{E}\left[\left|\int f_m \, \mathrm{d}\left(\tilde{\mu}_{t+1}^N - \tilde{T}\left(\tilde{\mu}_t^N, \tilde{\xi}_t\right)\right)\right|\right]$$
$$\leq \sup_{m \geq 1} \mathbb{E}\left[\mathbb{E}\left[\left|\int f_m \, \mathrm{d}\left(\tilde{\mu}_{t+1}^N - \tilde{T}\left(\tilde{\mu}_t^N, \tilde{\xi}_t\right)\right)\right| \, \middle| \, \tilde{x}_t^N, \tilde{\xi}_t\right]\right].$$

and obtain

$$
\mathbb{E}\left[\left|\left|\int f_m \, \mathrm{d}\left(\tilde{\mu}_{t+1}^N - \tilde{T}\left(\tilde{\mu}_t^N, \tilde{\xi}_t\right)\right)\right|\right|\, \Big|\, \tilde{x}_t^N, \tilde{\xi}_t\right]^2
$$

$$
= \mathbb{E}\left[\left|\left|\frac{1}{N}\sum_{i\in[N]}\left(f_m(x_{t+1}^i) - \mathbb{E}\left[f_m(x_{t+1}^i)\,\Big|\,\tilde{x}_t^N, \tilde{\xi}_t\right]\right)\right|\right|\, \Big|\, \tilde{x}_t^N, \tilde{\xi}_t\right]^2 \le \frac{4}{N} \to 0,
$$

while for the induction assumption we use the equicontinuous functions $\mu \mapsto \int f(\hat{T}(\mu,\xi))\hat{\pi}^\theta(\xi \mid \mu)\mathrm{d}\xi$ by Assumption 3 and Lemma 4. $\qquad\square$

## R  CONVERGENCE OF VALUE FUNCTION

*Proof of Lemma 6.* We show the required statement by first showing at all times $t$ that

$$
\sup_{f\in\mathcal{F}}\left|\mathbb{E}\left[f(\tilde{\mu}_t^N) - f(\hat{\mu}_t)\,\Big|\,\tilde{\mu}_0 = \mu = \hat{\mu}_0, \tilde{\xi}_0 = \xi = \xi_0\right]\right| \to 0. \tag{27}
$$

This is clear at time $t = 0$ by $\tilde{\mu}_0 = \mu = \hat{\mu}_0, \tilde{\xi}_0 = \xi = \xi_0$ and the weak LLN argument as in the proof of Lemma 5. At time $t = 1$, we analogously have

$$
\sup_{f\in\mathcal{F}}\left|\mathbb{E}\left[f(\tilde{\mu}_1^N) - f(\hat{\mu}_1)\,\Big|\,\tilde{\mu}_0 = \mu = \hat{\mu}_0, \tilde{\xi}_0 = \xi = \xi_0\right]\right|
$$

$$
\le \sup_{f\in\mathcal{F}}\left|\mathbb{E}\left[f(\tilde{\mu}_1^N) - f(\hat{T}(\tilde{\mu}_0^N, \tilde{\xi}_0))\,\Big|\,\tilde{\mu}_0 = \mu, \tilde{\xi}_0 = \xi\right]\right|
$$

$$
+ \sup_{f\in\mathcal{F}}\left|\mathbb{E}\left[f(\hat{T}(\tilde{\mu}_0^N, \tilde{\xi}_0)) - f(\hat{\mu}_1)\,\Big|\,\tilde{\mu}_0 = \mu = \hat{\mu}_0, \tilde{\xi}_0 = \xi = \xi_0\right]\right|,
$$

and

$$
\sup_{f\in\mathcal{F}}\left|\mathbb{E}\left[f(\tilde{\mu}_{t+1}^N) - f(\hat{\mu}_{t+1})\,\Big|\,\tilde{\mu}_0 = \mu = \hat{\mu}_0, \tilde{\xi}_0 = \xi = \xi_0\right]\right|
$$

$$
\le \sup_{f\in\mathcal{F}}\left|\mathbb{E}\left[f(\tilde{\mu}_{t+1}^N) - \int f(\hat{T}(\tilde{\mu}_t^N, \xi'))\hat{\pi}^\theta(\xi' \mid \tilde{\mu}_t^N)\mathrm{d}\xi'\,\Big|\,\tilde{\mu}_0 = \mu, \tilde{\xi}_0 = \xi\right]\right|
$$

$$
+ \sup_{f\in\mathcal{F}}\left|\mathbb{E}\left[\int f(\hat{T}(\tilde{\mu}_t^N, \xi'))\hat{\pi}^\theta(\xi' \mid \tilde{\mu}_t^N)\mathrm{d}\xi' - f(\hat{\mu}_{t+1})\,\Big|\,\tilde{\mu}_0 = \mu = \hat{\mu}_0, \tilde{\xi}_0 = \xi = \xi_0\right]\right|,
$$

for times $t + 1 \ge 1$, each with the weak LLN arguments applied to the **former** terms (conditioning not only on $\tilde{x}_t^N$, but also $\tilde{\xi}_t$), and the induction assumption applied to the **latter** terms, using the equicontinuous functions $\mu \mapsto \int f(\hat{T}(\mu,\xi))\hat{\pi}^\theta(\xi \mid \mu)\mathrm{d}\xi$ by Assumption 3 and Lemma 4. $\qquad\square$

## S  CONTINUITY OF VALUE FUNCTION

*Proof of Lemma 7.* For any $(\mu_n, \xi_n) \to (\mu, \xi)$, we show again by induction over all times $t$ that for any equicontinuous family $\mathcal{F}$,

$$
\sup_{f\in\mathcal{F}}\left|\mathbb{E}\left[f(\hat{\mu}_t) \mid \hat{\mu}_0 = \mu_n, \xi_0 = \xi_n\right] - \mathbb{E}\left[f(\hat{\mu}_t) \mid \hat{\mu}_0 = \mu, \xi_0 = \xi\right]\right| \to 0 \tag{28}
$$

as $N \to \infty$, from which the result follows. At time $t = 0$, we have by definition

$$
\sup_{f\in\mathcal{F}}\left|\mathbb{E}\left[f(\hat{\mu}_0) \mid \hat{\mu}_0 = \mu_n, \xi_0 = \xi_n\right] - \mathbb{E}\left[f(\hat{\mu}_0) \mid \hat{\mu}_0 = \mu, \xi_0 = \xi\right]\right| = 0.
$$

Analogously, at time $t = 1$ we have

$$
\sup_{f\in\mathcal{F}}\left|\mathbb{E}\left[f(\hat{\mu}_1) \mid \hat{\mu}_0 = \mu_n, \xi_0 = \xi_n\right] - \mathbb{E}\left[f(\hat{\mu}_1) \mid \hat{\mu}_0 = \mu, \xi_0 = \xi\right]\right|
$$

$$
= \sup_{f\in\mathcal{F}}\left|\mathbb{E}\left[f(\hat{T}(\hat{\mu}_0, \xi_0))\,\Big|\,\hat{\mu}_0 = \mu_n, \xi_0 = \xi_n\right] - \mathbb{E}\left[f(\hat{T}(\hat{\mu}_0, \xi_0))\,\Big|\,\hat{\mu}_0 = \mu, \xi_0 = \xi\right]\right|
$$

$$= \sup_{f \in \mathcal{F}} \left| f(\hat{T}(\hat{\mu}_n, \xi_n)) - f(\hat{T}(\hat{\mu}, \xi)) \right| \to 0$$

by equicontinuous $f$ and continuous $\hat{T}$ from Lemma 4.

Now assuming that (28) holds at time $t \geq 1$, then at time $t + 1$ we have

$$\sup_{f \in \mathcal{F}} \left| \mathbb{E}\left[ f(\hat{\mu}_{t+1}) \mid \hat{\mu}_0 = \mu_n, \xi_0 = \xi_n \right] - \mathbb{E}\left[ f(\hat{\mu}_{t+1}) \mid \hat{\mu}_0 = \mu, \xi_0 = \xi \right] \right|$$

$$= \sup_{f \in \mathcal{F}} \left| \mathbb{E}\left[ \int f(\hat{T}(\hat{\mu}_t, \xi'))\hat{\pi}^\theta(\xi' \mid \hat{\mu}_t)\mathrm{d}\xi' \,\middle|\, \hat{\mu}_0 = \mu_n, \xi_0 = \xi_n \right] \right.$$

$$\left. - \mathbb{E}\left[ \int f(\hat{T}(\hat{\mu}_t, \xi'))\hat{\pi}^\theta(\xi' \mid \hat{\mu}_t)\mathrm{d}\xi' \,\middle|\, \hat{\mu}_0 = \mu, \xi_0 = \xi \right] \right|$$

$$= \sup_{g \in \mathcal{G}} \left| \mathbb{E}\left[ g(\hat{\mu}_t) \mid \hat{\mu}_0 = \mu_n, \xi_0 = \xi_n \right] - \mathbb{E}\left[ g(\hat{\mu}_t) \mid \hat{\mu}_0 = \mu, \xi_0 = \xi \right] \right| \to 0$$

by induction assumption on equicontinuous functions $g \in \mathcal{G}$ by Assumptions 1a and 3, Lemma 4, and equicontinuous $f \in \mathcal{F}$, as in Theorem 1.

The convergence of $\left| Q^\theta(\mu_n, \xi_n) - Q^\theta(\mu, \xi) \right| \to 0$ thus follows by Assumption 1a. $\qquad\square$

