# OpenReview forum: "Learning Decentralized Partially Observable Mean Field Control for Artificial Collective Behavior"
_ICLR.cc/2024/Conference — ICLR 2024 poster_

### Official Review · Reviewer_xiBU · 2023-11-01

**Soundness:** 3 good
**Presentation:** 3 good
**Contribution:** 3 good
**Rating:** 6
**Confidence:** 2

**Summary:**

The manuscript introduces Decentralized Partially Observable Mean Field Control (Dec-POMFC) to extend scalable MFC to a decentralized and partially observable system. The paper includes rigorous theoretical proof. The experiments are performed on representative collective behavior tasks such as adapted Kuramoto and VIcsek swarming models, against the SoAT IPPO methods.

**Strengths:**

The paper tackles a challenging problem featuring partial observability, multiple agents, and decentralization. This area is worth exploring.

The paper is well-written and provides very details algorithm description and theorectical proof. The experiments are carefully carried on.

**Weaknesses:**

The experiments were carried out on adapted Kuramoto and VIcsek swarming models. The authors report training curves, and the returns as the number of agents increases in the main manuscript. However, there is no direct comparison with the SoAT methods in terms of the second method.  In addition, there is no explicit summary or discussion about significant improvement, and it is hard for the reviewer to judge the contribution of the proposed methods in terms of the experimental results.

**Questions:**

1. The manuscript mentioned extending the proposed methods to handle additional practical constraints and sparser interaction. However, the reviewer sees it can be hard given the adopted assumptions in the paper. The reviewer is curious about the feasibility of making this extension.

2. The author claims about the generality of the proposed method. Can the authors elaborate on which design in the algorithm contributes to this generality in addition to using RL itself?

---

> ### Author Response · Authors · 2023-11-17
> **Response to Reviewer xiBU**
>
> We thank the reviewer for constructive feedback on the experiments and their discussion, prospective theoretical extensions, and clarifying the generality of our approach. We have addressed the following in the revision manuscript:
>
> ---
>
> **W1: However, there is no direct comparison with the SotA methods in terms of the second method.**
>
> We completely agree with the reviewer's remarks and have added a comparison against SotA MAPPO (using a centralized critic) in Figure 4, which performs similar to IPPO. The comparison over number of agents $N$ was moved to Figures 23 and 24 (also for MAPPO).
>
> ---
>
> **W2: In addition, there is no explicit summary or discussion about significant improvement, and it is hard for the reviewer to judge the contribution of the proposed methods in terms of the experimental results.**
>
> We thank the reviewer for suggesting a more detailed discussion on improvements, which was added to the conclusion. Our method is of interest due to (i) its theoretical optimality guarantees while covering a large class of problems (see Q2 below), and (ii) its surprising simplicity in rigorously reducing complex Dec-POMDPs to MDPs, with the the same complexity as MDPs from fully observable MFC. We allow analyzing hard Dec-POMDPs via a tractable MDP. Despite its surprising simplicity, its performance often matches SotA methods.
>
> ---
>
> **Q1: The manuscript mentioned extending the proposed methods to handle additional practical constraints and sparser interaction. However, the reviewer sees it can be hard given the adopted assumptions in the paper. The reviewer is curious about the feasibility of making this extension.**
>
> For prospective extensions towards sparsity, we note that limits of dense graphs called graphons have been considered in literature for graphon MFGs [A] and MFC [B], while less dense graphs have been considered via Lp-graphons in MFGs [C]. An extension using sparse graphs either for MFC or also under partial observability remains to be executed, but we are optimistic that a combination of the above is feasible.
>
> [A] Caines, Peter E., and Minyi Huang. "Graphon mean field games and the GMFG equations: ε-Nash equilibria." 2019 IEEE 58th Conference on Decision and Control (CDC). IEEE, 2019.
>
> [B] Hu, Yuanquan, et al. "Graphon Mean-Field Control for Cooperative Multi-Agent Reinforcement Learning." Journal of the Franklin Institute (2023).
>
> [C] Fabian, Christian, Kai Cui, and Heinz Koeppl. "Learning sparse graphon mean field games." International Conference on Artificial Intelligence and Statistics. PMLR, 2023.
>
> ---
>
> **Q2: The author claims about the generality of the proposed method. Can the authors elaborate on which design in the algorithm contributes to this generality in addition to using RL itself?**
>
> We would like to underline the high generality of mean field games, which includes many real world system applications, e.g. finance and economic theory [D], engineering [E] or social networks [F]. While one essentially assumes exchangeable agents, note that exchangeable agents can still include heterogeneous classes of agents by including their class as part of the state [G, H]. Accordingly, our Dec-POMFC framework is also of high generality (covering a wide class of problems).
>
> [D] Carmona, Rene. "Applications of mean field games in financial engineering and economic theory." arXiv preprint arXiv:2012.05237 (2020).
>
> [E] Djehiche, Boualem, Alain Tcheukam, and Hamidou Tembine. "Mean-Field-Type Games in Engineering." AIMS Electronics and Electrical Engineering 1.1 (2017): 18-73.
>
> [F] Bauso, Dario, Hamidou Tembine, and Tamer Basar. "Opinion dynamics in social networks through mean-field games." SIAM Journal on Control and Optimization 54.6 (2016): 3225-3257.
>
> [G] Mondal, Washim Uddin, Vaneet Aggarwal, and Satish Ukkusuri. "Mean-Field Control based Approximation of Multi-Agent Reinforcement Learning in Presence of a Non-decomposable Shared Global State." Transactions on Machine Learning Research (2023).
>
> [H] Ganapathi Subramanian, Sriram, et al. "Multi Type Mean Field Reinforcement Learning." Proceedings of the 19th International Conference on Autonomous Agents and MultiAgent Systems. 2020.
>
> ---
>
> We hope that the above addresses the reviewer's concerns and are happy to provide further clarification if any concerns remain.

---

### Official Review · Reviewer_gUVQ · 2023-11-01

**Soundness:** 2 fair
**Presentation:** 2 fair
**Contribution:** 2 fair
**Rating:** 5
**Confidence:** 2

**Summary:**

This paper applies MARL to the proposed decentralized partially observable mean-filed control (Dec-POMFC) model, which can be reduced to single-agent MDP. A PPO-based method in the CTDE framework is applied to solve such problems. Theoretical analysis is also given.

**Strengths:**

- The paper is well written.

**Weaknesses:**

- It is unclear to me where the particular difficulty is from. If it is from the partial observation, how do you solve the partial observation problem? Note that many Dec-POMDP problems can be grouped into weakly-coupled POMDP [1] where the partial observation can be sufficient to make optimal decisions.
- The CTDE training is very common in MARL, and the baseline of IPPO is not fair as it cannot use global information while the proposed method uses more information.
- The benchmark looks simple. It would be good to include more realistic and complex benchmarks to indicate the importance of the studied question and the proposed method.

[1] Witwicki, Stefan, and Edmund Durfee. "Inﬂuence-Based Policy Abstraction for Weakly-Coupled Dec-POMDPs." *Proceedings of the international conference on automated planning and scheduling*. Vol. 20. 2010.

**Questions:**

- What is the particular difficulty of solving the Dec-POMFC system, and how does the proposed method solve such difficulty? It would be easier to follow the paper if these questions were explicitly explained.
- How much information do you assume each agent can observe? i.e. how severe is the partial observation problem in the proposed model and algorithm?

---

> ### Author Response · Authors · 2023-11-17
> **Response to Reviewer gUVQ**
>
> We thank the reviewer for constructive feedback on clarifying the main difficulties, relation to weakly-coupled POMDPs, comparisons against SotA, relevance of benchmarks, and the information available to agents. We have addressed the following in the revision manuscript:
>
> ---
>
> **W1a: It is unclear to me where the particular difficulty is from. If it is from the partial observation, how do you solve the partial observation problem?**
>
> **Q1: What is the particular difficulty of solving the Dec-POMFC system, and how does the proposed method solve such difficulty?**
>
> The main difficulties include rigorously reaching the Dec-POMFC and algorithmically solving its infinite-dimensional MDP, which are now explained at the start of Section 2:
>
> - Theoretical challenge: The original Dec-POMDP is hard, as each agent indeed has only partial information. It is *rewritten* into the Dec-POMFC, and we underline the theoretical contribution of *obtaining* the Dec-POMFC model, for which we develop a new theory for optimality of Dec-POMFC solutions in the hard finite Dec-POMDP (Theorems 1 and 2, and also the Lipschitz policy techniques in Appendix K).
> - Empirical challenge: As you indicate, the solution of Dec-POMFC itself remains hard, because its MDP state-actions are not just continuous, but infinite-dimensional. It is addressed by our algorithm using (i) kernel parametrizations and (ii) the rigorously justified policy gradients on the finite system instead of the commonly assumed infinite system (Theorem 3).
>
> ---
>
> **W1b: Note that many Dec-POMDP problems can be grouped into weakly-coupled POMDP [1] where the partial observation can be sufficient.**
>
> We thank the reviewer for pointing out possible important comparisons to existing POMDP techniques [A] to provide useful insight:
>
> - The Dec-POMFC model has a similar flavor to transition-decoupled POMDPs [A], as the mean field also abstracts influence from all other agents. However, both Dec-POMFC and [A] address different types of problems: [A] considers local per-agent states, while the mean field is both a globally, jointly defined state between all agents, and influenced by all agents (as a function of all agents' local states).
> - Furthermore, our framework is not a search framework [A], considers variable large numbers of agents, and solves for optimal policies depending on little information, allowing also for reactive policies (memory-less agents).
>
> [A] Witwicki, Stefan, and Edmund Durfee. "Inﬂuence-Based Policy Abstraction for Weakly-Coupled Dec-POMDPs." Proc. ICAPS. Vol. 20. 2010.
>
> ---
>
> **W2: The CTDE training is very common in MARL, and the baseline of IPPO is not fair.**
>
> We completely agree with the reviewer's remarks and have added a comparison against SotA MAPPO (centralized critic) in Figure 4, which performs similar to IPPO. The comparison over $N$ was moved to Figures 23 and 24 (also for MAPPO).
>
> ---
>
> **W3: The benchmark looks simple. It would be good to include more realistic and complex benchmarks to indicate the importance of the studied question and the proposed method.**
>
> We thank the reviewer for bringing up the topic of realistic benchmarks. We would like to respond that the Vicsek and Kuramoto models are realistic for many applications, see e.g. biological or artificial swarming systems [B], robotic swarm coordination [C], or Kuramoto power grids and neural networks [D]. We agree that more complex agent models exist, but the main challenge is the size of the multi-agent system, and less the complexity of a single agent.
>
> [B] Vicsek, Tamás, et al. "Application of statistical mechanics to collective motion in biology." Physica A: Statistical Mechanics and its Applications 274.1-2 (1999): 182-189.
>
> [C] Virágh, Csaba, et al. "Flocking algorithm for autonomous flying robots." Bioinspiration & Biomimetics 9.2 (2014): 025012.
>
> [D] Rodrigues, Francisco A., et al. "The Kuramoto model in complex networks." Physics Reports 610 (2016): 1-98.
>
> ---
>
> **Q2: How much information do you assume each agent can observe? i.e. how severe is the partial observation problem in the proposed model and algorithm?**
>
> - In our model, Eq. (1), we assume agents observe arbitrarily little information, so long as observations depend only on the agent state and mean field. Therefore, our framework handles quite general types of observations.
> - The result naturally depends on the observation model. For example, if Vicsek agents cannot observe absolute or relative headings, they cannot possibly align. Action distributions of agents differ only depending on their observations. Nonetheless, our theory allows solving for an optimal shared policy under such observations, which due to limited information remains unable to align properly even if optimized.
>
> ---
>
> In case the above sufficiently addresses the reviewer's concerns, we would be happy if the reviewer considers increasing their score. We are ready to provide further clarification should any concerns remain.

---

> > ### Comment · Reviewer_gUVQ · 2023-11-20
> > **Reply to authors**
> >
> > I thank the authors for your further elaboration. Now I think I have a better understanding of the contribution of this paper, a simplified model for infinite agents with partial observation under certain assumptions, i.e. Dec-POMFC. Additional optimality guarantee is analyzed. I appreciate the authors for making me understand it.
> >
> > From the algorithmic perspective of MARL, I insist that this paper doesn't show much interesting stuff. The algorithm is not new, and the performance is flat.
> >
> > However, I am not an MFC guy, as other reviewers suggested, maybe guys from MFC would like to see such a reduced model and want to make more assumptions and analysis based on this paper. I will increase my score to be marginal.

---

### Official Review · Reviewer_PQjP · 2023-11-01

**Soundness:** 3 good
**Presentation:** 3 good
**Contribution:** 3 good
**Rating:** 8
**Confidence:** 3

**Summary:**

This paper studies mean field control problems with decentralized decisions and partial observations. After introducing the problem, the authors prove theoretical results connecting the mean field problem and the N-agent problem. They then propose a policy-gradient based method and provide experimental results on several examples.

**Strengths:**

The paper seems rigorous and has both theoretical contributions and numerical experiments.

**Weaknesses:**

Some assumptions seem quite restrictive, such as Assumption 1b which says that the policies should be uniformly Lipschitz (see question below).

**Questions:**

Is it possible to replace the assumption on policies (which means restricting a priori the set of policies) by an assumption on the model which would ensure that the optimal policy satisfies this Lipschitz property? And would this be sufficient for your purposes

---

> ### Author Response · Authors · 2023-11-17
> **Response to Reviewer PQjP**
>
> We thank the reviewer for their positive evaluation and insightful feedback on weakening the assumptions. We have addressed the following in the revision manuscript:
>
> ---
>
> **W1: Some assumptions seem quite restrictive, such as Assumption 1b which says that the policies should be uniformly Lipschitz (see question below).**
>
> **Q1: Is it possible to replace the assumption on policies (which means restricting a priori the set of policies) by an assumption on the model which would ensure that the optimal policy satisfies this Lipschitz property? And would this be sufficient for your purposes?**
>
> - We are very thankful for the insightful remark. As an alternative, our results indeed generalize to arbitrary policies, if the state space is finite and observations of agents do not depend on the mean field. But there is a trade-off, as the alternative assumptions instead restrict the class of solvable Dec-POMDPs. We have added the alternative to Assm. 1b.
> - We also underline that Assm. 1b is less empirically restrictive, as it is always fulfilled for discrete observations, and suffices for our algorithms, which optimize over such Lipschitz kernel-based classes of policies.
>
> ---
>
> We hope that the above addresses the reviewer's concerns and are happy to provide further clarification if any concerns remain.

---

### Author Response · Authors · 2023-11-17
**Response to all reviewers**

We would like to thank all of the reviewers for their time and constructive feedback.  We have uploaded a new revision, and the major changes are summarized as follows:

- We have added additional experimental comparisons to more state-of-the-art MARL methods (MAPPO, Figures 3 and 24).
- We have added discussions on (i) relation to existing Dec-POMDP techniques (introduction, before contribution), (ii) central difficulties (beginning of Section 2), and (iii) significant improvement or advantages of our contributions (in conclusion).
- We have significantly relaxed Assm. 1b (Lipschitz policies) in the finite state and local observation setting, thanks to the idea of reviewer PQjP.
- To make space, the description of problems in the main text was shortened.

---

### Meta-Review · Area_Chair_Tdqe · 2023-12-24

**Metareview:**

### Summary
The paper studies MARL via mean field control (MFC), as it offers a potential solution to scalability. While prior work fails to consider decentralized and partially observable systems, this paper extends MFCs and it introduces Decentralized Partially Observable Mean Field Control (Dec-POMFC) to extend scalable MFC to a decentralized and partially observable system

###  Strengths
+ Well written and motivated  presentation of Dec-MFCs, explanation of challenges and experimental evaluations.
+ Detailed algorithm description and formal proof

### Weaknesses:
- lack of SOTA baselines and discussion
- Restrictive assumptions, limiting practical application of theoretical insights.

**Justification For Why Not Higher Score:**

The paper draft needs was initially lacking, and needed several improvements.

**Justification For Why Not Lower Score:**

The paper presents an interesting problem setup for cooperative multi-agent learning. All reviewers agree with the utility and novelty of the proposed setup.

---

### Decision · Program_Chairs · 2024-01-16

Accept (poster)